# The mechanical origin of snow avalanche dynamics and flow regime transitions

Xingyue Li[1], Betty Sovilla[2], Chenfanfu Jiang[3], and Johan Gaume[1,2]

[1]School of Architecture, Civil and Environmental Engineering, Swiss Federal Institute of Technology, Lausanne, Switzerland
[2]WSL Institute for Snow and Avalanche Research, SLF, Davos, Switzerland
[3]Computer and Information Science Department, University of Pennsylvania, Philadelphia, USA

**Correspondence:** Johan Gaume (johan.gaume@epfl.ch)

**Abstract.** Snow avalanches cause fatalities and economic damages. Key to their mitigation entails the understanding of snow avalanche dynamics. This study investigates the dynamic behavior of snow avalanches, using the Material Point Method (MPM) and an elastoplastic constitutive law for porous cohesive materials. By virtue of the hybrid Eulerian-Lagrangian nature of MPM, we can handle processes involving large deformations, collisions and fractures. Meanwhile, the elastoplastic model enables us to capture the mixed-mode failure of snow, including tensile, shear and compressive failure. Using the proposed numerical approach, distinct behaviors of snow avalanches, from fluid-like to solid-like, are examined with varied snow mechanical properties. In particular, four flow regimes reported from real observations are identified, namely, cold dense, warm shear, warm plug and sliding slab regimes. Moreover, notable surges and roll-waves are observed peculiarly for flows in transition from cold dense to warm shear regimes. Each of the flow regimes shows unique flow characteristics in terms of the evolution of the avalanche front, the free surface shape, and the vertical velocity profile. We further explore the influence of slope geometry on the behavior of snow avalanches, including the effect of slope angle and path length on the maximum flow velocity, the runout angle and the deposit height. Unified trends are obtained between the normalized maximum flow velocity and the scaled runout angle as well as the scaled deposit height, reflecting analogous rules with different geometry conditions of the slope. It is found that the maximum flow velocity is mainly controlled by the friction between the bed and the flow, the geometry of the slope, and the snow properties. We reveal the crucial effect of both flow and deposition behaviors on the runout angle. Furthermore, our MPM modeling is calibrated and tested with simulations of real snow avalanches. The evolution of the avalanche front position and velocity from the MPM modeling shows reasonable agreement with the measurement data from literature. The MPM approach serves as a novel and promising tool to offer systematic and quantitative analysis for mitigation of gravitational hazards like snow avalanches.

## 1 Introduction

Snow avalanches have long been threatening infrastructures and human lives. Buildings, roads and railways can be severely damaged, causing profound economic losses. Moreover, the fatalities induced by snow avalanches are significant, which are about 100 annually in the European Alps during the last four decades (Techel et al., 2016). Due to climate change, the frequency

and risk of snow avalanches are still increasing (Choubin et al., 2019). It is thus of great importance to mitigate snow avalanche hazards, which highly relies on the understanding of their complex dynamic behavior.

Snow can behave as a fluid or as a solid under different conditions, leading to distinct behaviors of snow avalanches in reality (Gaume et al., 2011; Ancey, 2016). Characterizing different flow regimes of snow avalanches has played a significant role in hazard mapping and design of mitigation measures (Gauer et al., 2008). Traditionally, two flow regimes were considered for snow avalanches, namely, dense snow avalanches and powder snow avalanches. A recent study by Köhler et al. (2018) highlighted the role of snow temperature in classifying the flow regimes and extended the traditional classification. The starting, flowing and stopping signatures of snow avalanches were used to distinguish seven flow regimes, including four dense flow regimes, two powder flow regimes, and a snowball flow regime. Although the flow regimes have been identified based on macro flow behavior from the real measurements, their underlying physics remains unclear. Numerical and theoretical models can provide efficient and comprehensive analysis to shed light on the internal flow characteristics underpinning the macro flow behavior, but are extremely challenging (Faug et al., 2018). To date, there has been no recognized model capable of capturing and analyzing the diverse behaviors of snow avalanches in a systematic and well-controlled way. Furthermore, the crucial effects of snow mechanical property and terrain geometry on snow avalanche dynamics have been widely recognized, but are sparsely investigated due to practical challenges. Only limited numerical and real-measurement studies were reported (Keshari et al., 2010; Fischer et al., 2012, 2015; Steinkogler et al., 2014).

Popular classical numerical tools for modeling snow avalanches primarily apply two-dimensional (2D) depth-averaged methods based on shallow water theory (Naaim et al., 2013; Rauter et al., 2018), which fail to capture important flow characteristics along the surface-normal direction such as velocity distribution (Eglit et al., 2020). Nevertheless, 2D models are computationally efficient and provide acceptable accuracy, which serve as a powerful tool in many applications like hazard mapping. In comparison, three-dimensional (3D) simulations can fully resolve flow variations in all dimensions, which consequently require longer computation time. In recent years, particle-based continuum methods, including Smooth Particle Hydrodynamics (SPH), Particle Finite Element Method (PFEM), and Material Point Method (MPM), have gained increasing popularity in avalanche modeling, as they are able to easily handle large deformations and discontinuities (Abdelrazek et al., 2014; Salazar et al., 2016; Gaume et al., 2018a). In particular, MPM has proven to be an effective and efficient tool in investigating snow (Stomakhin et al., 2013; Gaume et al., 2018a, b, 2019). Compared with SPH where boundary conditions are challenging to generalize, MPM can readily address complex boundaries (Raymond et al., 2018). Moreover, MPM does not suffer from the time consuming neighbor searching that is inevitable in many mesh-free approaches like SPH (Mast et al., 2014). Both PFEM and MPM use a set of Lagrangian particles and a background mesh to solve mass and momentum conservation of a system. In contrast to PFEM, each particle in MPM has fixed mass, as it allows to naturally guarantee mass conservation. However, the fixed mass meanwhile leads to difficulty in adding or removing particles from the system (Larsson et al., 2020). The computational cost of MPM is lower than that of PFEM according to simulations with same formulation (Papakrivopoulos, 2018).

This study applies MPM in 2D (slope-parallel and slope-normal) to explore the distinct behaviors of snow avalanches and the key controlling factors of snow avalanche dynamics. To facilitate efficient computation and capture important flow features along the surface-normal direction, our 2D MPM modeling neglects variations along the flow width. MPM is a hybrid Eulerian-

Lagrangian approach, which uses Lagrangian particles to track mass, momentum and deformation gradient, and adopts Eulerian

background mesh to solve and update the motion of the particles. By virtue of the hybrid Eulerian-Lagrangian nature of MPM, processes with large deformations, fractures, collisions and impacts can be well simulated (Mast et al., 2014; Gaume et al., 2018a, b, 2019). In addition, continuous solid-fluid phase transition and coexistence of solid-like and fluid-like behaviors can be captured with implementation of proper constitutive models (Stomakhin et al., 2013; Gaume et al., 2018a). MPM has been increasingly adopted to investigate gravity-driven flows like landslides, debris flows and avalanches (Soga et al., 2016; Abe and

Konagai, 2016; Gaume et al., 2018a). This study will highlight the capability of MPM in capturing different flow regimes of snow avalanches from fluid-like shear flow to solid-like sliding slab, by adopting a finite strain elastoplastic constitutive model. Furthermore, it will be demonstrated that the proposed numerical approach serves as a promising tool to systematically study the key influencing factors of snow avalanche dynamics, including snow mechanical property and slope geometry.

## 2    Methodology

### 2.1    The Material Point Method (MPM)

Assuming a continuous material, MPM discretizes it into Lagrangian particles (material points) to trace mass, momentum and deformation gradient, and adopts Eulerian grids to solve the motion of the particles and update their states. In particular, the particle motion is governed by mass and momentum conservation as follows

$$\frac{D\rho}{Dt} + \rho \nabla \cdot \boldsymbol{v} = 0 \tag{1}$$

$$\rho \frac{D\boldsymbol{v}}{Dt} = \nabla \cdot \boldsymbol{\sigma} + \rho \mathbf{g} \tag{2}$$

where $\rho$ is density, $t$ is time, $\boldsymbol{v}$ is velocity, $\boldsymbol{\sigma}$ is the Cauchy stress, $\boldsymbol{g}$ is the gravitational acceleration. As the mass carried by each particle does not vary, the balance of mass is satisfied naturally. The momentum balance is solved with a regular background Eulerian grid mesh and the discretization of the weak form of Eq. (2). The explicit MPM algorithm by Stomakhin

et al. (2013) is applied with a symplectic Euler time integrator. Details of the adopted MPM time stepping algorithm can be found in Stomakhin et al. (2013); Jiang et al. (2016); Gaume et al. (2018a). Note compared to Gaume et al. (2018a), this study uses Affine Particle-In-Cell (APIC) method (Jiang et al., 2015), by which angular momentum is preserved in addition to linear momentum.

MPM relies on a continuum description and requires a constitutive model for the considered material. The Cauchy stress $\boldsymbol{\sigma}$

in Eq. (2) is related to the strain thorough an elastoplastic constitutive law as follows

$$\boldsymbol{\sigma} = \frac{1}{J} \frac{\partial \Psi}{\partial \boldsymbol{F}_E} \boldsymbol{F}_E^T \tag{3}$$

where $\Psi$ is the elastoplastic potential energy density, $\boldsymbol{F}_E$ is the elastic part of the deformation gradient $\boldsymbol{F}$, and $J = \det(\boldsymbol{F})$. Note that various constitutive models can be implemented into the framework of MPM to capture different materials and

their distinct behaviors. For example, a non-associated Mohr-Coulomb model was applied to model landslide and dam failure (Zabala and Alonso, 2011; Soga et al., 2016), and a non-associated Drucker-Prager model was used to simulate sand (Klár et al., 2016). In this study, we use the associated Modified Cam Clay model developed for snow (Gaume et al., 2018a), which reproduces mixed-mode snow fracture and compaction hardening.

## 2.2 Finite strain elastoplastic model

The elastoplastic model in this study is borrowed from Gaume et al. (2018a), which consists of a mixed-mode shear-compression yield surface, a hardening law, and an associative flow rule. We recall the main characteristics of the three key components. On the basis of laboratory experiments (Reiweger et al., 2015) and simulations based on X-ray computed tomography (Hagenmuller et al., 2015; Chandel et al., 2015; Srivastava et al., 2017), the yield surface is defined in the space of the $p - q$ invariants of the stress tensor as follows

$$y(p,q) = (1+2\beta)q^2 + M^2(p+\beta p_0)(p-p_0) \tag{4}$$

$p$ is the pressure calculated as $p = -\text{tr}(\boldsymbol{\tau})/d$, where $\boldsymbol{\tau}$ is the Kirchhoff stress tensor and $d$ is the dimension. $q$ is the Mises stress defined as $q = (3/2\, \mathbf{s} : \mathbf{s})^{1/2}$, where $\mathbf{s} = \boldsymbol{\tau} + p\mathbf{I}$ is the deviatoric stress tensor and $\mathbf{I}$ is the identity matrix. $p_0$ is the consolidation pressure and denotes the isotropic compressive strength. $\beta p_0$ is the isotropic tensile strength, where $\beta$ reflects the cohesion. $M$ is the slope of the critical state line, which characterizes the internal friction.

When the $p - q$ state of the material is inside the yield surface (i.e. $y(p,q) < 0$), the material behaves elastically and follows Hooke's law (St Venant-Kirchhoff Hencky strain). Plastic behavior happens if $y(p,q) = 0$. Depending on the volumetric plastic strain $\epsilon_v^p$, hardening or softening is implemented by expanding or shrinking the yield surface according to the following hardening law

$$p_0 = K\sinh(\xi \max(-\epsilon_v^p, 0)) \tag{5}$$

where $K$ is the bulk modulus and $\xi$ is the hardening factor. Under compression ($\dot{\epsilon}_v^p < 0$), $p_0$ increases, leading to hardening and promoting compaction. Under tension ($\dot{\epsilon}_v^p > 0$), $p_0$ decreases, resulting in softening and allowing fracture.

A flow rule needs to be adopted when plastic behavior occurs. Referring to Gaume et al. (2018a), this study uses an associative plastic flow rule reported by Simo (1992) and Simo and Meschke (1993). The applied flow rule follows the principle of maximum plastic dissipation, which maximizes the rate of plastic dissipation. It is worth noting that the second law of thermodynamics is fully satisfied by using the plastic model with the flow rule. More details can be found in Gaume et al. (2018a).

# 3 Snow avalanches on ideal slopes

## 3.1 Model setup

To examine the behavior of snow avalanches under a well-controlled condition, the setup with a rectangular snow sample and an ideal slope is adopted as illustrated in Fig. 1. The snow sample is initially placed at the top of the slope, and will flow

down under gravity. The inclined slope is connected to the horizontal ground using a circular arc with a central angle equaling to the slope angle $\theta$. The arc and the horizontal ground are named connecting arc zone and deposition zone in the following discussion, respectively. To investigate different flow regimes of snow avalanches, the properties of snow are systematically

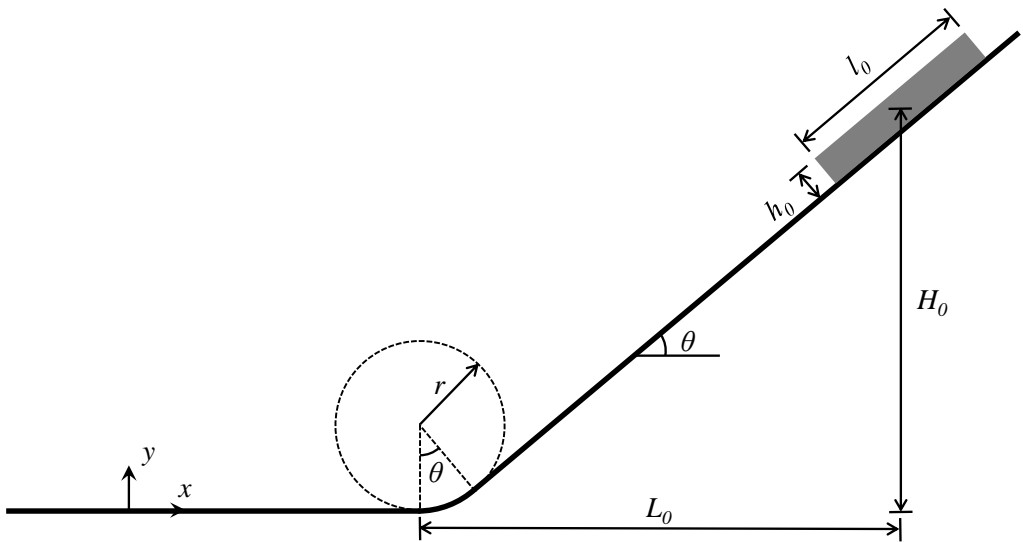

**Figure 1.** Model setup for MPM modeling of snow avalanches on an ideal slope.

varied, including the friction coefficient $M$, the tension/compression ratio $\beta$, the hardening factor $\xi$, and the initial consolidation pressure $p_0^{ini}$. In addition, the effect of slope angle $\theta$ and horizontal length $L_0$ in Fig. 1 is studied with five groups of simulations

as summarized in Table 1. For each group, the snow properties are changed within the prescribed ranges. Groups I, II, III are conducted to study the influence of slope angle $\theta$, whilst Groups II, IV, V are designed to examine the effect of the horizontal length $L_0$. When $\theta$ is varied in Groups I, II, III, the horizontal length $L_0$ is fixed and the drop height $H_0$ is adjusted as listed in Table 1. Alternatively, one could fix the drop height $H_0$ and change the horizontal length $L_0$. Note $L_0/h_0$, $L_0/l_0$, and $L_0/r$ are kept constant when $L_0$ is varied in Groups II, IV, V, by changing $h_0$, $l_0$ and $r$ accordingly. An increased $L_0$ leads to the scale-up

of the setup, resulting in the rise of the drop height $H_0$. Detailed parameters adopted in the MPM simulations are summarized in Table 1. The size of the background Eulerian mesh in MPM is selected to be small enough to guarantee the simulation

accuracy and resolution, and meanwhile be large enough to shorten the computation time. The time step is constrained by the CFL condition and the elastic wave speed to secure the simulation stability. The simulation data are exported every 1/24 s.

**Table 1.** Parameters adopted in the MPM simulations of snow avalanches on ideal slopes.

|  |  | Group I | Group II | Group III | Group IV | Group V |
|---|---|---|---|---|---|---|
| **Snow** | Density $\rho$ (kg/m$^{-3}$) | 250 | 250 | 250 | 250 | 250 |
|  | Young's modulus $E$ (MPa) | 3 | 3 | 3 | 3 | 3 |
|  | Poisson's ratio $\nu$ | 0.3 | 0.3 | 0.3 | 0.3 | 0.3 |
|  | Friction coefficient $M$ [*] | 0.1~1.5 | 0.1~1.5 | 0.1~1.5 | 0.1~1.5 | 0.1~1.5 |
|  | Tension/compression ratio $\beta$ [*] | 0.0~1 | 0.0~1 | 0.0~1 | 0.0~1 | 0.0~1 |
|  | Hardening factor $\xi$ [*] | 0.1~10 | 0.1~10 | 0.1~10 | 0.1~10 | 0.1~10 |
|  | Initial consolidation pressure $p_0^{ini}$ (kPa) [*] | 3~30 | 3~30 | 3~30 | 3~30 | 3~30 |
|  | Initial height $h_0$ (m) | 2 | 2 | 2 | 5 | 8 |
|  | Initial length $l_0$ (m) | 12 | 12 | 12 | 30 | 48 |
| **Slope** | Bed friction coefficient $\mu$ | 0.5 | 0.5 | 0.5 | 0.5 | 0.5 |
|  | Slope angle $\theta$ ($^\circ$) | 30 | 40 | 50 | 40 | 40 |
|  | Radius $r$ (m) | 10 | 10 | 10 | 25 | 40 |
|  | Drop height $H_0$ (m) | 37.1 | 52.8 | 73.5 | 132.0 | 211.2 |
|  | Horizontal length $L_0$ (m) | 65 | 65 | 65 | 162.5 | 260 |
| **Simulation control** | Mesh size (m) | 0.05 | 0.05 | 0.05 | 0.05 | 0.05 |
|  | Time step (s) | $2.3 \times 10^{-4}$ | $2.3 \times 10^{-4}$ | $2.3 \times 10^{-4}$ | $2.3 \times 10^{-4}$ | $2.3 \times 10^{-4}$ |
|  | Frame rate (FPS) | 24 | 24 | 24 | 24 | 24 |

[*] $M$ values include 0.1, 0.5, 1.0, 1.5. $\beta$ values include 0.0, 0.3, 0.6, 1.0. $\xi$ values include 0.1, 0.5, 1.0, 10.0. $p_0^{ini}$ values include 3 kPa, 12 kPa, 21 kPa, 30 kPa.

## 3.2   Typical flow regimes

In each group of our MPM simulations, four typical flow regimes are captured with the changing mechanical properties of snow. Fig. 2(a) shows four representative cases in Group II, where distinct flow regimes are observed with the different snow properties summarized in Table 2. From top to bottom, we observe Regime 1 to Regime 4. The flow in Regime 1 behaves as a fluid or a dry cohesionless granular flow, whose free surface is continuous. Since the height of the flow in Regime 1 is excessively small compared with the others, it is scaled up to be three times higher along the bed normal direction for better visualization in Fig. 2(a). Small surges are observed especially at the front of the flow in Regime 1. The flow in Regime 2 demonstrates a more fluctuated free surface and a discontinuous tail, due to the occurrence of a granulation process. The flow height of the granular flow is higher compared with that of the flow in Regime 1, since the granules can be notably accumulated in the connecting arc and deposition zones. The flow in Regime 3 demonstrates ductile behavior, and slides down the slope and reaches the horizontal deposition zone with no significant deformation and no cracks. In contrast, clear cracks and broken

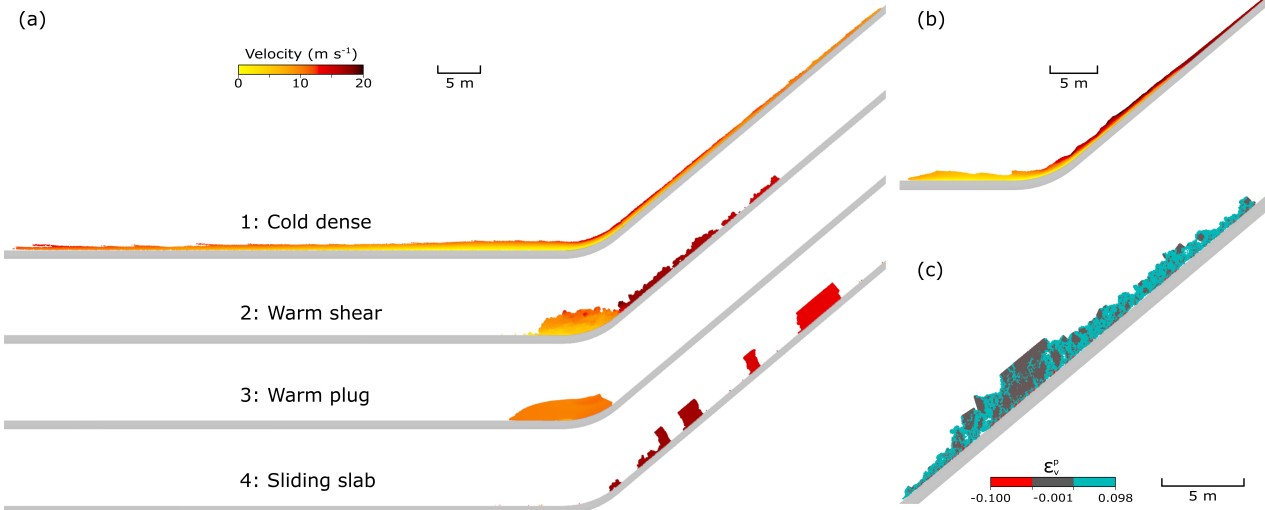

**Figure 2.** (a) Flows in four typical flow regimes captured at t = 8.3 s in the MPM simulations (Table 1, Group II). From top to bottom: cold dense, warm shear, warm plug and sliding slab. The cold dense flow is scaled up to be three times higher along the bed normal direction for better visualization. (b) A flow with surges and small granules in transition from cold dense regime to warm shear regime at t = 8.3 s. The color denotes velocity as in (a). (c) The early stage (t = 5.5 s) of the warm shear flow in (a). Videos of the simulations are provided as supplements.

pieces are noticed in the flow in Regime 4. The initial snow sample in Regime 4 breaks into multiple blocks shortly after its release from the slope.

The four flow regimes in Fig. 2(a) show similar features with the identified flow regimes by Köhler et al. (2018) based on the real measurements and observations, namely, cold dense, warm shear, warm plug, and sliding slab regimes. The detailed macro flow characteristics and internal shearing behavior of the flows will be expatiated in the following section. The information of the energy of the flows is provided in Appendix A. In addition, flows in transition between the flow regimes can be captured as well. For example, Fig. 2(b) shows a flow in transition from cold dense to warm shear flow regimes, as it demonstrates the characteristics of both regimes. Significant surges and roll-waves occur in Fig. 2(b), showing similarity with the cold dense flow. On the other hand, small granules are observed at the early stage of the flow (see supplementary video 4), presenting characteristics of the warm shear flow. This transitional flow is modeled with $M\beta = 0.15$ ($M = 0.5$; $\beta = 0.3$), whose properties are in between of the cold dense and warm shear flows listed in Table 2. Although only one flow regime is characterized for each of the flows in Fig. 2(a), it is worth noting that a single snow avalanche may display different flow regimes at its different locations and at different instants (Kern et al., 2009). Fig. 2(c) shows the warm shear flow in Fig. 2(a) at its early stage (t = 5.5 s), where the red and blue materials are respectively in compression and tension and the dark gray denotes the initial state of the material. The red particles are hardly visible as they are located at the bottom layer of the flow, which indicates that the snow inside the core of the avalanche is mainly under tension or at the initial state. At t = 5.5 s, the flow demonstrates the

characteristics of sliding slab, as the dark gray part slides along the slope and is seldom sheared. The initial sliding slab in Fig. 2(c) can indeed transform into the warm shear flow in Fig. 2(a) after it reaches the deposition zone.

**Table 2.** Snow properties adopted in the MPM modeling of the flows in the four typical flow regimes.

| Flow regime | $M$ | $\beta$ | $\xi$ | $p_0^{ini}$ (kPa) | $\beta p_0^{ini}$ (kPa) | $M\beta$ |
|---|---|---|---|---|---|---|
| Cold dense | 0.1 | 0 | 1 | 3 | 0 | 0 |
| Warm shear | 1.5 | 0.3 | 10 | 30 | 9 | 0.45 |
| Warm plug | 0.5 | 1 | 0.1 | 12 | 12 | 0.5 |
| Sliding slab | 1.5 | 1 | 1 | 21 | 21 | 1.5 |

All the demonstrated flows in the four typical regimes share identical initial and boundary conditions except for the snow properties. From the simulations, it is clear that higher $M$ and $\beta$ gives a more frictional and cohesive flow, since they reflect the internal friction and cohesion, respectively. For instance, the $M$ and $\beta$ of the flow in the warm shear regime are higher than that of the cold dense regime, which facilitate the formation of the granules and the higher flow height. The hardening coefficient $\xi$ and the initial consolidation pressure $p_0^{ini}$ also affect the flow behavior, whose influence depends on the $M$ and $\beta$ according to our sensitivity study. As listed in Table 2, the tensile strength $\beta p_0^{ini}$ and $M\beta$ consistently increase from the cold dense to the sliding slab flow regimes, which give indications on the possible underpinning factors controlling the transition of the flow regimes.

### 3.2.1 Front evolution

Fig. 3 illustrates the evolution of the front position and velocity for the flows in the four typical flow regimes in Fig. 2(a). In some of the simulated flows, scattered particles are observed at the flow front (i.e. the warm shear flow and sliding slab flow in supplementary video 2), which need to be excluded in the determination of the front position as they separate from the main body of the flow. Hence, the front position is determined by ruling out 1% of the particles at the front of the flow. The front in Fig. 3(a) is calculated as the horizontal distance between the current front position and the initial front position. The gray band in Fig. 3(a) shows the region where a flow front enters the connecting arc zone, below and above which the flow front is on the slope and in the horizontal deposition area, respectively. The evolution of front velocity in Fig. 3(b) is plotted with two constant velocities $v_{max}^b$ and $v_{max}^f$, which are the theoretical velocities of a sliding rigid block with and without consideration of bed friction, respectively. Referring to Fig. 1, if a rigid block slides down the slope, its path length prior to the connecting arc zone is $l = L_0/\cos\theta - r\tan\theta - 0.5l_0$. Its acceleration along the flow direction is $a^b = g(\sin\theta - \mu\cos\theta)$ considering gravity and friction or $a^f = g\sin\theta$ with a frictionless bed, where $\mu$ is the bed friction coefficient fixed to 0.5 as listed in Table 1. Given $l$, $a^b$, $a^f$, the theoretical velocities when the block goes to the end of the slope can be calculated as $v_{max}^b = \sqrt{2a^b l}$ and $v_{max}^f = \sqrt{2a^f l}$ with and without consideration of the bed friction, respectively.

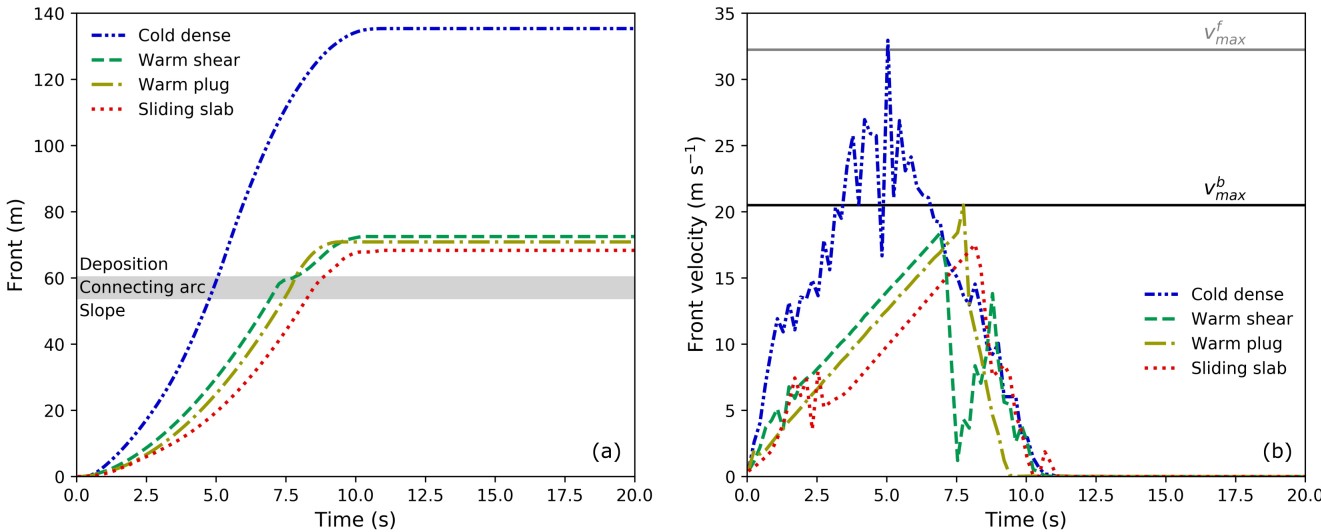

**Figure 3.** Evolution of (a) front and (b) front velocity of the flows in the four typical regimes (Table 1, Group II).

When the flows are on the slope (front < 53.82 m), the front of the cold dense flow is the fastest, followed by the warm shear and warm plug flows, and the sliding slab is the slowest. Indeed, the front velocity in Fig. 3(b) generally gives a consistent trend. The warm shear and sliding slab flows demonstrate fluctuations at around 1.2 s and 2.2 s in Fig. 3(b), due to the breakage of the snow sample. The tensile strength of the snow in the warm shear flow (9 kPa) is smaller than the sliding slab (21 kPa), leading to the earlier breakage and earlier fluctuations in Fig. 3(b). The fluctuations in the front velocity of the cold dense flow

last longer, which might be due to the turbulence and surges as observed in Fig. 2(a). After the flows enter the connecting arc zone (53.82 m < front < 60.25 m), the fronts of the cold dense, warm plug and sliding slab flows evolve smoothly in Fig. 3(a), whilst the front of the warm shear flow is sharply slowed down at the end of the connecting arc zone before it goes to the horizontal deposition zone. In the warm shear flow, discrete granules form a discontinuous flow front. After the scattered granules arrive at the end of the connecting arc zone, they quickly stop, leading to the stagnancy in the increase of the front

position until the arrival of continuously incoming granules. Indeed, the warm shear flow in Fig. 3(b) shows the sharp velocity reduction at around 7.5 s, after which the growth of velocity occurs thanks to the incoming flow. A significant drop is also observed from the front velocity of the cold dense flow at around 4.8 s in Fig. 3(b), corresponding well to the moment that the flow front reaches the connecting arc zone in Fig. 3(a). Indeed, this reduction is mainly contributed from the changed slope geometry in addition to the turbulence and surges in the cold dense flow. In Fig. 3(b), the maximum front velocities of the warm

shear, warm plug and sliding slab are close to $v_{max}^b$, whilst the maximum front velocity of the cold dense flow reaches $v_{max}^f$. This indicates different dominant factors of the maximum flow velocity. The maximum velocity of the flows in warm shear, warm plug and sliding slab regimes is chiefly governed by the frictional behavior between the flow and the bed. In contrast, the maximum velocity of the cold dense flow is mainly controlled by the snow properties, where the low friction and low cohesion facilitate a higher velocity. When the flow fronts enter the deposition zone (front > 60.25 m), all the flows start to slow down

gradually. It is noticed that the front velocity of the warm shear flow shows fluctuations from around 7.5 s, which are chiefly because of the discrete nature of the snow granules at the flow front (see supplementary video 1). As the front velocity of the warm shear flow decreases at around 7.5 s, the warm plug flow exceeds the warm shear flow. Nevertheless, the final front of the warm shear flow goes further as it stops later. The final fronts of the four flows show a consistent relation as that obtained when they are on the slope. Before the flows stop, the decelerations of the fronts (slope of velocity in Fig. 3(b)) are similar, which might be governed by bed friction.

### 3.2.2 Evolution of free surface shape and vertical velocity profile

Fig. 4 shows the evolution of free surface and velocity profile of the flows in the different flow regimes. The velocity profile is obtained at $x = 50$ m. The free surface of the cold dense flow in Fig. 4(a) is scaled up 15 times along the bed-normal direction to visualize the fluctuations at the surface. The height of the cold dense flow is much smaller than the initial flow height, since it is highly sheared throughout its flow depth as shown in Fig. 4(e). The velocity profiles in Fig. 4(e) are smooth, indicating the continuous shearing along the flow depth. The shape of the velocity profile in Fig. 4(e) does not change much with time, whilst the flow speed and the shear rate decrease as the flow tends to stop. Generally speaking, the cold dense flow behaves as a fluid or a noncohesive granular flow, in agreement with the characterization by Köhler et al. (2018). The warm shear flow in Fig. 4(b) demonstrates a fluctuated free surface because of the formed granules. Correspondingly, its velocity profile shows fluctuations as well. As illustrated in Fig. 4(f), the warm shear flow is fully sheared along the flow depth direction before it stops. Moreover, its flow depth can exceed the initial flow height due to the piling up and accumulation of snow granules. The shear behavior and the piling up feature are indeed consistent with the identified warm shear regime by Köhler et al. (2018). Nevertheless, instead of a noncohesive granular flow characterized by Köhler et al. (2018), the flow in our MPM modeling does have cohesion (see Table 2), which helps the formation of the granules. The warm plug flow remains a block and is seldom sheared when it slides on the slope. Upon its arrival at the connecting arc zone, significant shearing occurs due to the changed shape of the connecting arc zone. As shown in Figs 4(c) and (g), the front of the warm plug flow is notably sheared at t = 8.0 s, the flow body is only sheared at the bottom layer at t = 8.3 s, and the flow tail is seldomly sheared at t = 8.8 s. The sliding slab in Fig. 4(d) shows the sliding down of the slabs from t = 9.0 s to 9.2 s and the accumulated slabs in the connecting arc and deposition zones at t = 10.5 s. As there are particles stopping on the slope, the tail of the free surface collapses onto the slope. The shearing behavior inside the slabs is extremely limited as shown in Fig. 4(h). Both the warm plug and the sliding slab behave as solid-like objects, while the snow of the sliding slab flow is more brittle and produces slab fractures.

### 3.3 Effect of slope angle and path length on flow dynamics

### 3.3.1 Maximum velocity and deposition height

Figs 5 and 6 demonstrate the evolution of the maximum avalanche velocity on the slope $v_{max}$ with the normalized avalanche deposit height $h_d/h_0$, under the effects of the slope angle $\theta$ and the horizontal length $L_0$ (reflecting the path length), respectively. The deposit height $h_d$ is defined as the maximum avalanche height along the bed normal direction after snow avalanches

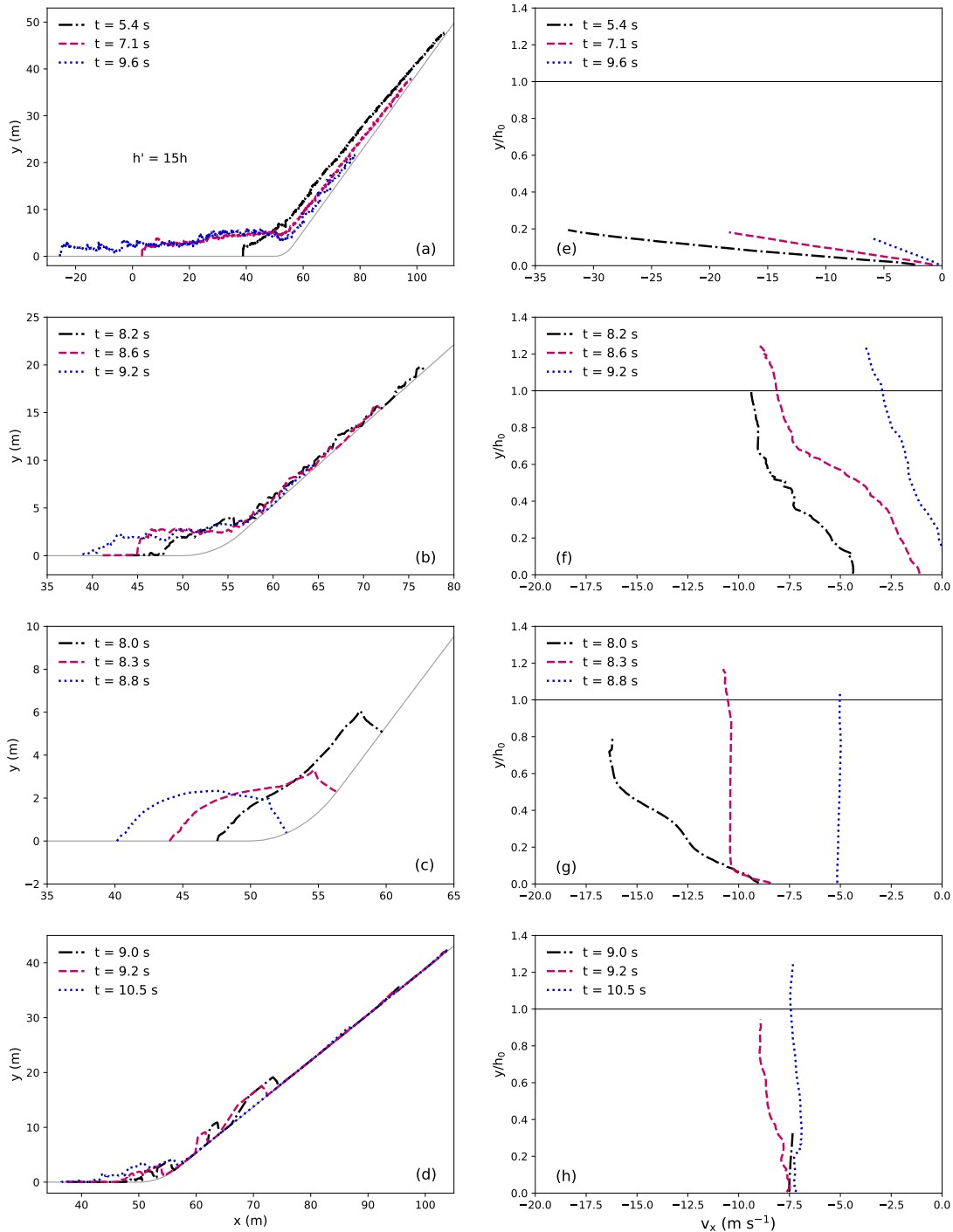

**Figure 4.** Evolution of free surface shape (left column) and velocity profile at $x = 50$ m (right column) for the flows in the different regimes (Table 1, Group II). (a)&(e) cold dense; (b)&(f) warm shear; (c)&(g) warm plug; (d)&(h) sliding slab. The free surface of the cold dense flow in (a) is scaled up 15 times along the bed normal direction for better visualization.

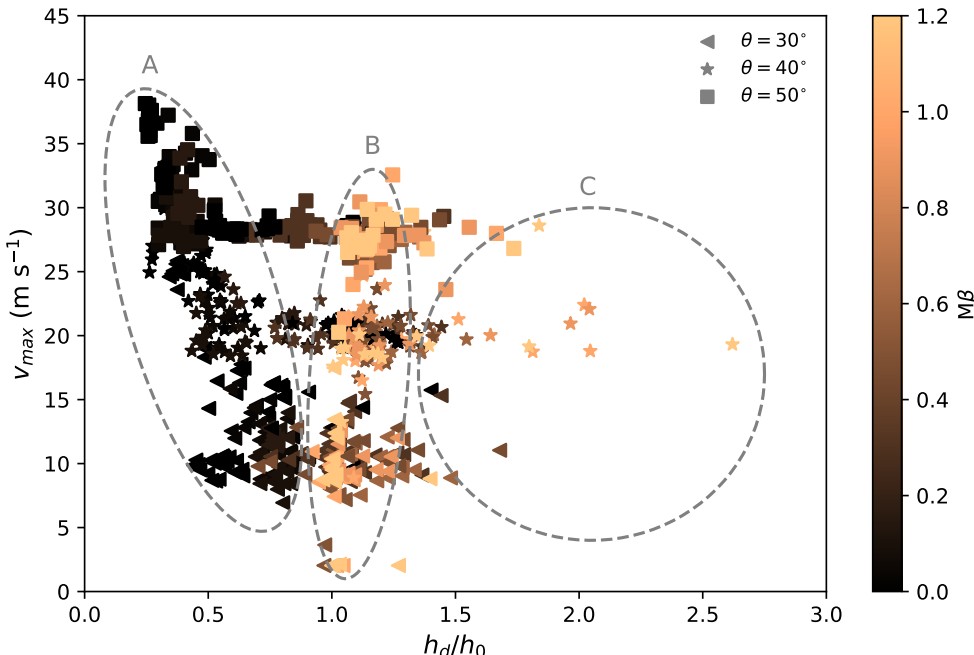

**Figure 5.** Evolution of the maximum velocity with the normalized deposit height for varying slope angles $\theta$.

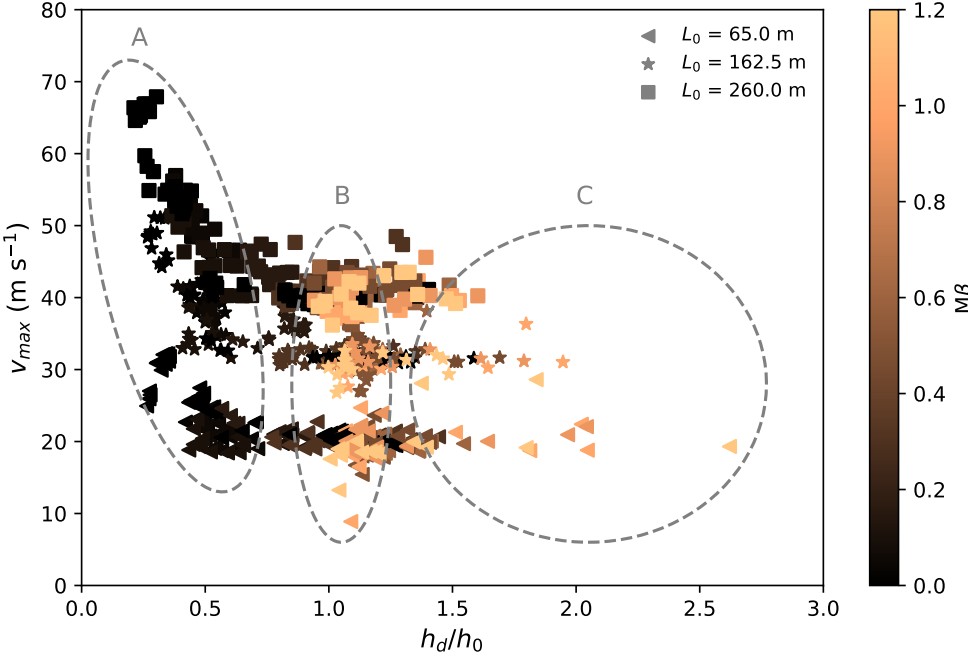

**Figure 6.** Evolution of the maximum velocity with the normalized deposit height for different horizontal lengths $L_0$.

stop. The maximum velocity of snow avalanches is usually obtained before their arrival at the deposition zone. To analyze our MPM data with theoretical predictions, the maximum velocity $v_{max}$ in Figs 5 and 6 is determined when the flow is on the slope. For all the simulated groups in Figs 5 and 6, a similar trend is observed, with three zones indicating different flow characteristics and flow regimes. Zone A shows a maximum velocity which tends to be negatively correlated with the deposit height. A typical flow regime is the cold dense regime, in which a higher maximum flow velocity leads to a longer run-out distance and a smaller deposit height. The black data in Zone A reflects small snow friction and cohesion, which agrees with the snow properties of the cold dense flow. Zone B has a deposit height close to the initial height of the snow sample. This characteristic is normally captured from the warm plug and sliding slab flow regimes. Note that, in addition to the typical case in the sliding slab regime in Fig. 4(d) which demonstrates accumulated snow in the deposition zone, there are other cases with slabs sliding down the slope and stopping in the deposition zone without piling up and snow accumulation. These cases in the sliding slab regime give a final deposit height close to the initial flow height. The high snow friction and cohesion reflected by the light color in Zone B indeed indicate the snow properties of the warm plug and sliding slab flows. In Zone C, the deposit height is notably larger than the initial height. In this case, representative flow regimes are the warm shear flow and the sliding slab flow, where the accumulation of snow can be significant after the flows deposit. It is found that the flowing and deposition behaviors of snow avalanches are primarily controlled by the snow friction and snow cohesion ($M$ and $\beta$), as we observe the clear transition of colors denoting $M\beta$ in the different zones in Figs 5 and 6. The scattered colors of some points, such as the dark points in Zone C, indicate the additional effects of snow brittleness (reflected by $\xi$) and snow compressive strength ($p_0$).

Slope angle is a key factor in evaluating the trigger, flow and deposition of snow avalanches (Gaume, 2012; Sovilla et al., 2010). Fig. 5 shows the positive correlation between the slope angle $\theta$ and the maximum velocity on the slope $v_{max}$. When $\theta$ is varied with a fixed $L_0$ (see Fig. 1), the drop height $H_0$ is increased accordingly, which gives a larger initial potential energy of the flow and consequently a higher $v_{max}$. The effect of increased path length reflected by $L_0$ is similar to the outcome of the growth of $\theta$, as shown in Fig. 6. It is interesting to observe the similar trend for the different groups of simulations with varying $\theta$ and $L_0$, which hints an analogous physical rule behind the trend. Indeed, a unified relation can be obtained as shown in Fig. 7, by scaling $v_{max}$ and $h_d$ as follows

$$v_{max}^* = \frac{v_{max} - v_{max}^b[1 - e^{-M(1+\beta)/\kappa_1}]}{v_{max}^f} \tag{6}$$

$$h^* = \frac{h_d}{h_0}[1 - e^{-M(1+\beta)\kappa_2/l}] \tag{7}$$

The normalization of $v_{max}$ takes into account $v_{max}^b$ and $v_{max}^f$, which are the theoretical predictions with a frictional bed and a frictionless bed, respectively. The consideration of $v_{max}^b$ and $v_{max}^f$ reflects the influence of bed friction, slope angle and path length. In addition, the effect of snow friction ($M$) and cohesion ($\beta$) is also considered. The deposit height $h_d$ is scaled with the initial height of the snow slab, the snow properties, and the path length. Note two constant coefficients $\kappa_1$ and $\kappa_2$ are used to account for other possible factors including snow compressive strength and brittleness, where $\kappa_1$ is dimensionless and $\kappa_2$ has a dimension of length. In this study, $\kappa_1$ and $\kappa_2$ are 0.2 and 200 m, respectively. According to Eq. (6), when the friction $M$

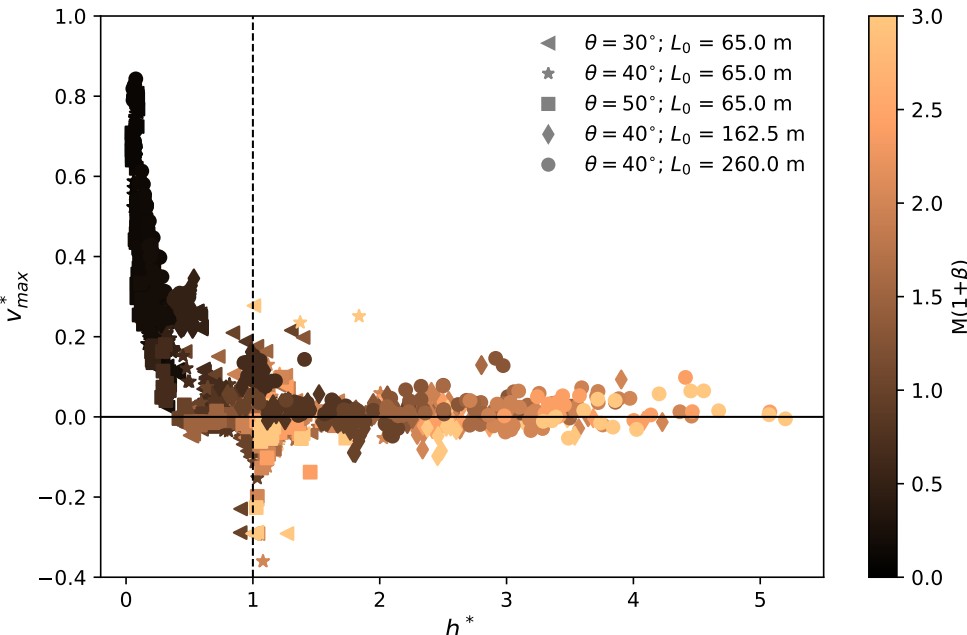

**Figure 7.** Unified relation between the scaled maximum velocity and the scaled deposit height.

and cohesion $\beta$ of snow are high, the numerator is close to $v_{max} - v_{max}^b$, and a zero numerator indicates a maximum velocity close to the theoretical prediction considering a rigid block sliding on a frictional bed. As shown in Fig. 7, the data around the zero line hint that the maximum velocity of the flows are chiefly controlled by the friction between the flow and the bed. On the other hand, when $M$ and $\beta$ tend to zero, $v_{max}^*$ approaches $v_{max}/v_{max}^f$ in Eq. (6), reflecting how close is the maximum flow velocity to the theoretical prediction with a rigid block sliding on a frictionless bed. Correspondingly, the cases with small $M$ and $\beta$ in Fig. 7 reflect a maximum flow velocity primarily governed by snow properties, instead of the frictional behavior between the flow and the bed. A representative case is the cold dense flow in Fig. 3(b), where its maximum velocity is close to $v_{max}^f$ as the flow is highly sheared. Furthermore, data below the zero line is observed in Fig. 7, corresponding to the cases where the snow box either stays on the slope with limited displacement or slides down the slope with a velocity sometimes decreased (i.e. not a constant acceleration as assumed in the calculation of $v_{max}^b$).

### 3.3.2 Maximum velocity and $\alpha$

The runout angle $\alpha$ is defined as $\alpha = \arctan(H/L)$. $H$ and $L$ are total vertical drop and total horizontal reach, respectively, determined based on the top point of the release zone and the front of the final deposit (Lied and Bakkehøi, 1980). Figs 8 and 9 show the relation between $v_{max}$ and $\alpha$, including MPM data and real-measurement data collected from McClung and Gauer (2018). For the five groups of MPM simulations varying the slope angle $\theta$ and horizontal length $L_0$, all of them largely follow a two-stage relation between $v_{max}$ and $\alpha$: an initially decreasing $v_{max}$ and a subsequently constant $v_{max}$ with the increase of $\alpha$.

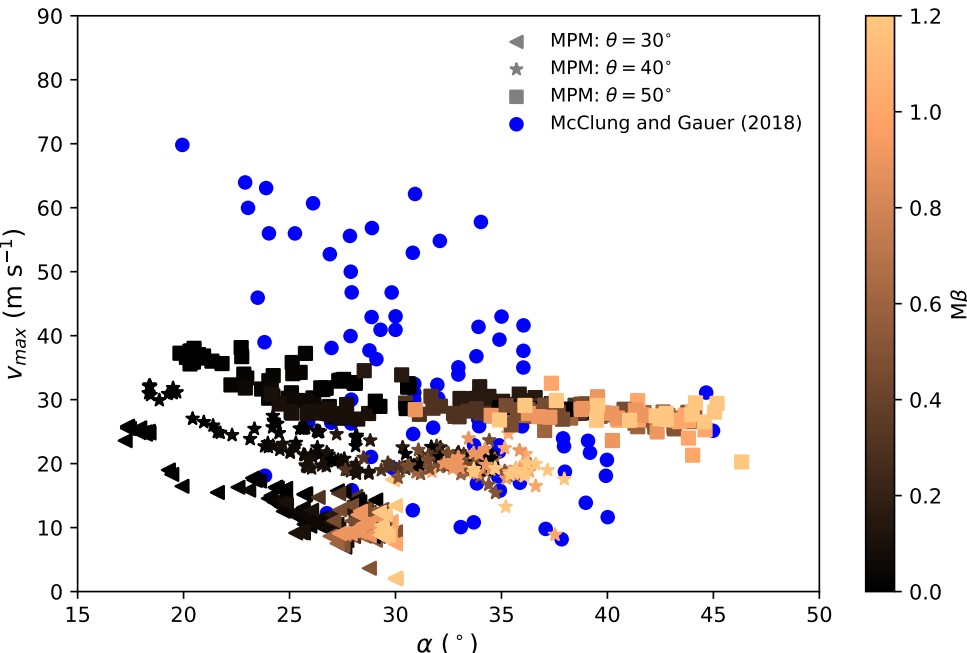

**Figure 8.** Evolution of the maximum velocity with $\alpha$ for varying slope angles $\theta$.

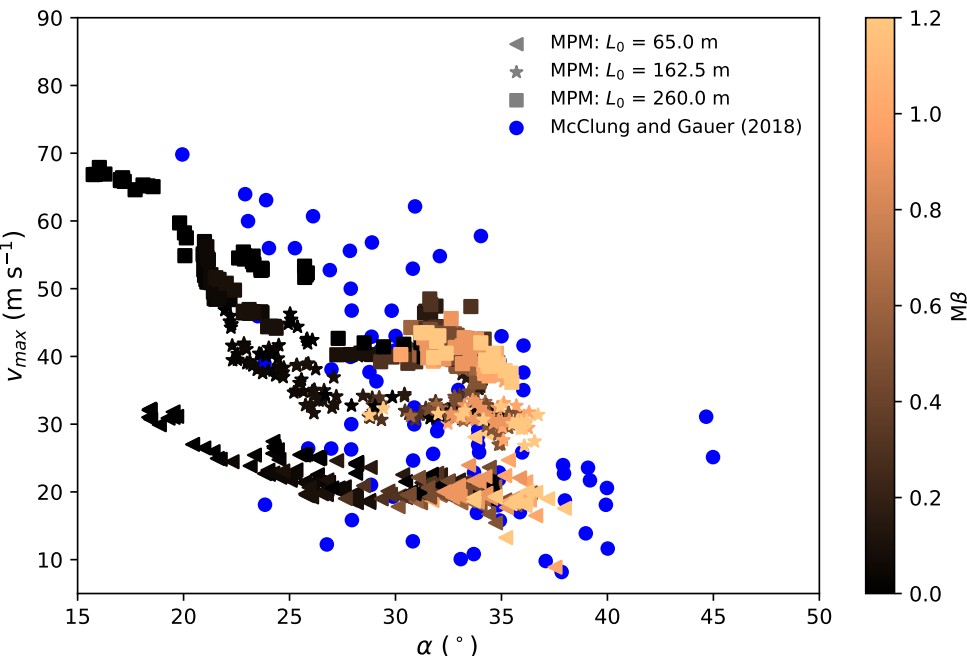

**Figure 9.** Evolution of the maximum velocity with $\alpha$ for different horizontal lengths $L_0$.

As demonstrated in Figs 8 and 9, the first stage mainly consists of cases with low friction and cohesion, whilst the second stage is chiefly composed of cases with high friction and cohesion. At the first stage, a higher $v_{max}$ leads to a longer runout distance and thus a smaller $\alpha$. This indicates the dominant effect of $v_{max}$ in controlling the runout distance, which might be due to the positive correlation between the velocity and the kinetic energy of a snow avalanche. Indeed, it has been recognized that the runout distance is tightly related to the kinetic energy of the flow upon its arrival at the deposition zone (Sovilla et al., 2006). Obviously, the dominance of $v_{max}$ disappears at the second stage, as a similar $v_{max}$ gives a notably different $\alpha$. The runout distance at this stage is mainly affected by the deposition behavior, instead of the flowing behavior. For example, assuming a warm plug flow and a warm shear flow sharing an identical $v_{max}$ before they reach the deposition zone, their runout distances can differ much, since the warm plug flow may stop abruptly whilst the warm shear flow may gradually become steady and have a relatively longer runout distance.

From Figs 8 and 9, the effects of $\theta$ and $L_0$ on $v_{max}$ are similar, as both of them have a positive correlation with $v_{max}$. In addition, the slope angle $\theta$ also influences the maximum runout angle as shown in Fig. 8. The larger the slope angle, the larger the maximum runout angle. This is due to the definition of the runout angle $\alpha$, which gives a maximum runout angle close to the slope angle $\theta$. The maximum runout angle is reached when a flow stops on the slope with the modeled configuration. With $\theta = 30°$, several flows stay on the slope and have $\alpha \approx \theta$. All the flows with $\theta = 40°$ and $50°$ go to the connecting arc and deposition zones, giving $\alpha < \theta$. Note that the runout angle $\alpha$ has been correlated to the mean slope angle $\beta$ in exploring the runout distance of snow avalanches (Lied and Bakkehøi, 1980; Barbolini et al., 2000; Delparte et al., 2008). As ideal slopes are adopted (see Fig. 1) here, the mean slope angle $\beta$ is close to the slope angle $\theta$. Indeed, the positive correlation between the maximum runout angle and the slope angle in Fig. 8 agrees with the $\alpha - \beta$ model (Lied and Bakkehøi, 1980). The MPM results in Figs 8 and 9 generally fall onto the range of the real-measurement data from McClung and Gauer (2018). In particular, the lower-bound of $v_{max}$ from the real measurements is recovered with the MPM simulation. Note the case with $v_{max} = 70$ m/s serving as the upper-bound of the field data was a powder snow avalanche, whose behavior differs much from the simulated dense snow avalanches. In addition, the path length of the upper-bound case is significantly higher than the adopted ones in the MPM simulations (McClung and Gauer, 2018). This upper-bound case can indeed be captured with our MPM modeling by varying the model setup, but is not the focus here. It was reported by McClung and Gauer (2018) that the runout angle has a negative correlation with the maximum front velocity, but with wide scatter as observed from the blue dots in Fig. 8 or Fig. 9. According to our sensitivity study, the scatter might be a result of different terrain conditions (e.g. slope angle), release positions (e.g. path length), and snow properties. In addition, some data might be on the plateau stage where the runout distance is governed by the deposition behavior instead of the maximum front velocity.

Fig. 10 demonstrates a unified trend with the dimensionless velocity $v_{max}^*$ in Eq. (6) and $\alpha^*$ as follows

$$\alpha^* = \frac{\alpha}{\alpha^b} \tag{8}$$

where $\alpha^b$ is calculated by assuming a sliding rigid block. Referring to Fig. 1, the velocity of the block increases from 0 to $\sqrt{2a^b l}$ as it slides down from the upstream to the end of the frictional slope. With an assumption that the velocity of the block does not change before and after it goes across the connecting arc zone, its runout distance on the deposition zone can be calculated, with

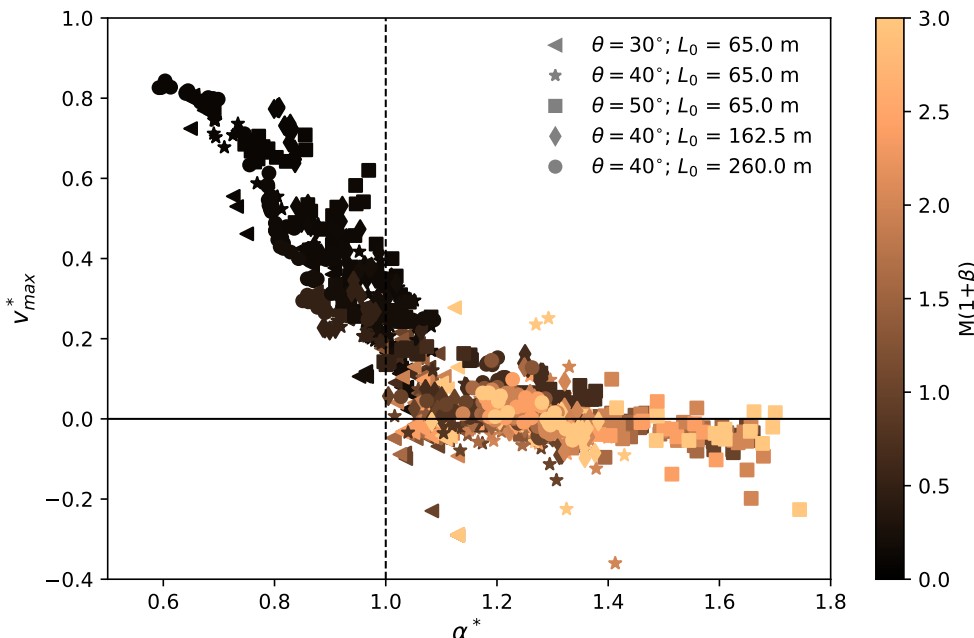

**Figure 10.** Unified relation between the scaled maximum velocity and the scaled $\alpha$.

an initial velocity of $\sqrt{2a^b l}$, a constant acceleration of $-\mu g$, and a final velocity of 0. $\alpha^b$ can then be derived as $\alpha^b = \arctan\mu$. It is interesting to obtain $\alpha^b$ solely dependent on the bed friction coefficient $\mu$. The scaled runout angle $\alpha^* = 1$ means a runout distance fully consistent with the prediction using the sliding rigid block theory, whilst $\alpha^* < 1$ and $\alpha^* > 1$ denote a runout distance which is longer and shorter than the predicted one with the sliding rigid block theory, respectively. Indeed, the data with $\alpha^* < 1$ in Fig. 10 generally have low friction and cohesion, which reasonably produce the longer runout distances. On the contrary, the cases with $\alpha^* > 1$ are typically more frictional and cohesive, leading to the shorter runout distances. Note that the data close to the zero line in Fig. 10 correspond to the cases at the plateau stage in Figs 8 and 9. As discussed in Fig. 7, when the friction $M$ and cohesion $\beta$ are high, a zero $v^*_{max}$ comes from a maximum velocity $v_{max}$ that approaches the theoretical prediction $v^b_{max}$ with consideration of a rigid block sliding on a frictional bed, indicating the maximum velocity $v_{max}$ is dominated by the frictional behavior between the flow and the bed. On the other hand, the $v_{max}$ of the cases far above and below the zero line in Fig. 10 is governed by the snow properties.

## 4 Snow avalanches on irregular terrains

To testify the capability of the MPM modeling in capturing key dynamic features (i.e. front velocity and position) of snow avalanches, five reported real avalanches with different complex terrains are simulated. All the cases are modelled in 2D, neglecting the variation along the flow width direction. The adopted geometry of the terrains is borrowed from the literatures.

As no detailed snow properties of the avalanches were measured and reported, the applied snow properties in MPM refer to the description of the snow type and snow condition. In particular, three of the avalanches mainly consisted of dry, loose and new snow, whilst the other two were chiefly composed of wind packed and settled old snow. Correspondingly, two groups of snow properties are adopted as summarized in Table 3. Based on the determined snow densities ($150\,\text{kg/m}^{-3}$ for new snow and $250\,\text{kg/m}^{-3}$ for old snow), the Young's modulus and tensile strength can be estimated using the relations from Gaume (2012)

and Scapozza (2004). The friction of the slope is calibrated according to the existing data of the real avalanches. Figs 11-15 show the MPM simulation results in comparison with the reported data from the literatures. Particularly, the evolution of the scaled front velocity is examined along the flow path (Gauer, 2014). The front velocity from the field was obtained by means of Doppler radar devices and photo analyses. Different measurement approaches may give different velocities, but are generally consistent with one another (Rammer et al., 2007). The comparison basis between velocities from numerical modeling and real

measurements needs to be carefully checked, as it is sometimes questionable (Fischer et al., 2014; Rauter and Köhler, 2020). For example, depth-averaged velocities from numerical modeling cannot be directly compared to peak intensity velocities from Droppler radar measurements (Rauter and Köhler, 2020). In Figs 11-15, the front velocity from MPM is determined as the approach velocity (Rauter and Köhler, 2020), which is calculated from the evolution of the front position with time and is assumed to be comparable with the data from the different measurement techniques. Note that this approach velocity has

a different definition from the velocity of the particles at the front of a flow, although their values are almost identical in our simulations. The geometry of the terrain is denoted by the gray dash curves in Figs 11-15, where the coordinates $x$ and $y$ are normalized with the vertical drop height $H_0$. The red bands in Figs 11-14 denote measurement error.

**Table 3.** Adopted parameters in the five MPM simulations of snow avalanches on real terrains.

| | | Case I | Case II | Case III | Case IV | Case V |
|---|---|---|---|---|---|---|
| **Snow** | Density $\rho$ (kg/m$^{-3}$) | 150 | 250 | 150 | 250 | 150 |
| | Young's modulus $E$ (MPa) | 0.47 | 6.45 | 0.47 | 6.45 | 0.47 |
| | Poisson's ratio $\nu$ | 0.3 | 0.3 | 0.3 | 0.3 | 0.3 |
| | Friction coefficient $M$ | 0.8 | 0.8 | 0.8 | 0.8 | 0.8 |
| | Tension/compression ratio $\beta$ | 0.1 | 0.2 | 0.1 | 0.2 | 0.1 |
| | Hardening factor $\xi$ | 0.5 | 0.5 | 0.5 | 0.5 | 0.5 |
| | Initial consolidation pressure $p_0^{ini}$ (kPa) | 10 | 20 | 10 | 20 | 10 |
| | Initial tensile strength $\beta p_0^{ini}$ (kPa) | 1 | 4 | 1 | 4 | 1 |
| **Slope** | Bed friction coefficient $\mu$ [*] | 0.46 | 0.46 | 0.63 | 0.51 | 0.46 |
| Simulation control | Mesh size (m) | 0.05 | 0.05 | 0.05 | 0.05 | 0.05 |
| | Time step (s) | $2.3 \times 10^{-4}$ | $2.3 \times 10^{-4}$ | $2.3 \times 10^{-4}$ | $2.3 \times 10^{-4}$ | $2.3 \times 10^{-4}$ |
| | Frame rate (FPS) | 24 | 24 | 24 | 24 | 24 |

[*] Calibrated parameter.

Case I and II in Figs 11 and 12 are two avalanches successively released at the north-west flank of the Weissfluh-Northridge (Gubler et al., 1986; Gauer, 2014), whose velocity was measured with continuous wave (CW) Doppler-radar. The snow forming the first avalanche was dry mostly loose new snow, which produced a powder cloud. In comparison, the second avalanche consisted of wind packed snow, which led to blocky slab-type release. It is noticed that the consistency between the MPM results and the measured data is better for the second avalanche in Fig. 12. The underestimated maximum front velocity in Fig. 11 might be due to the challenge of capturing the powder cloud of the first avalanche with MPM. The front velocity of a powder snow avalanche is normally obtained from the frontal dilute part, whose velocity can be higher than the dense core of the avalanche (Sovilla et al., 2015). In addition, the neglection of entrainment in the simplified MPM simulation may also contribute to the discrepancy in Fig. 11. It is suspected that the first release induced much more entrainment than the second one, considering the availability of the snow to be entrained.

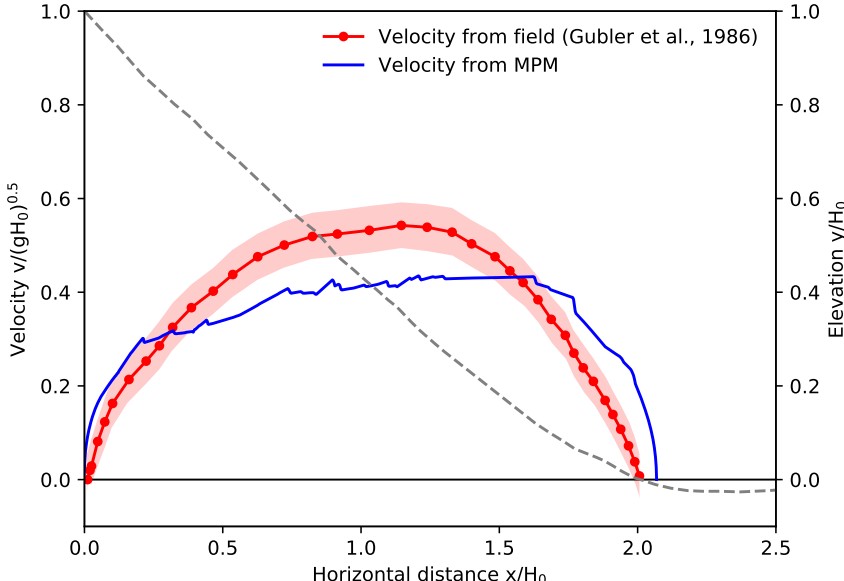

**Figure 11.** Front velocity distribution along the flow path for Case I: Weissfluh-Northridge 1982-03-12 a1 (Davos, Switzerland). Drop height $H_0 = 236$ m.

Fig. 13 shows the avalanche of Case III, happened after strong snowfall (Gauer, 2014). There were no field observations due to the stormy weather. The velocity was measured by a pulsed Doppler-radar. The snow was conjectured to be dry or only slightly moist. The adopted snow properties in the MPM modeling refer to that of new snow, which are assumed to be identical to the snow in Case I as listed in Table 3. Fig. 13 illustrates reasonable agreement between the MPM and the measured data, in terms of both the final front position and the maximum front velocity. During the flowing process, MPM tends to underestimate the front velocity, which might be related to the dry nature of the snow as discussed in Case I (Fig. 11). Compared with Cases I and II, the front velocity evolution of Case III is more fluctuated, as the terrain is more irregular.

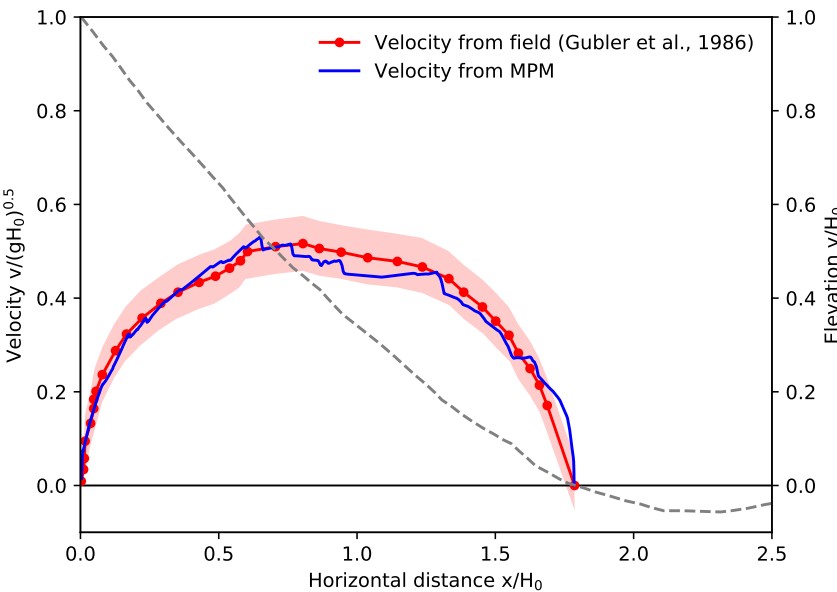

**Figure 12.** Front velocity distribution along the flow path for Case II: Weissfluh-Northridge 1982-03-12 a2 (Davos, Switzerland). Drop height $H_0 = 177$ m.

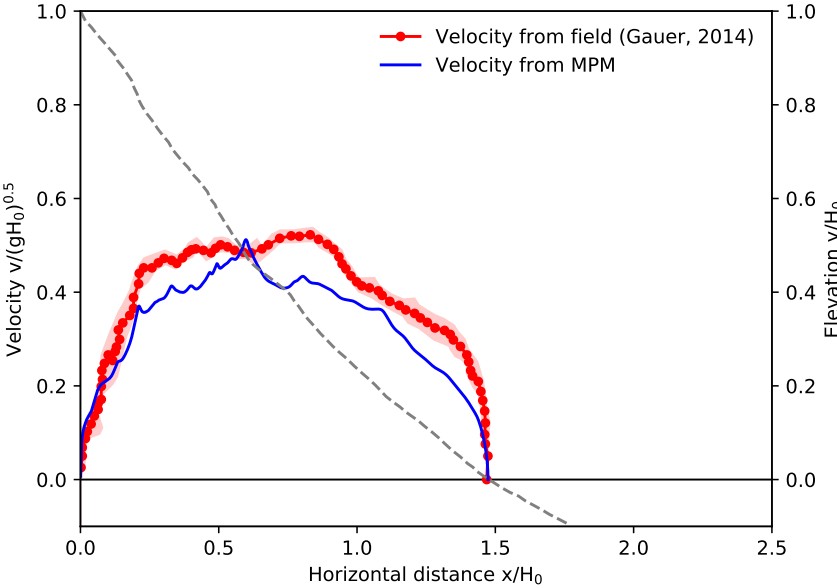

**Figure 13.** Front velocity distribution along the flow path for Case III: Himmelegg 1990-02-14 (Alberg, Austria). Drop height $H_0 = 352$ m.

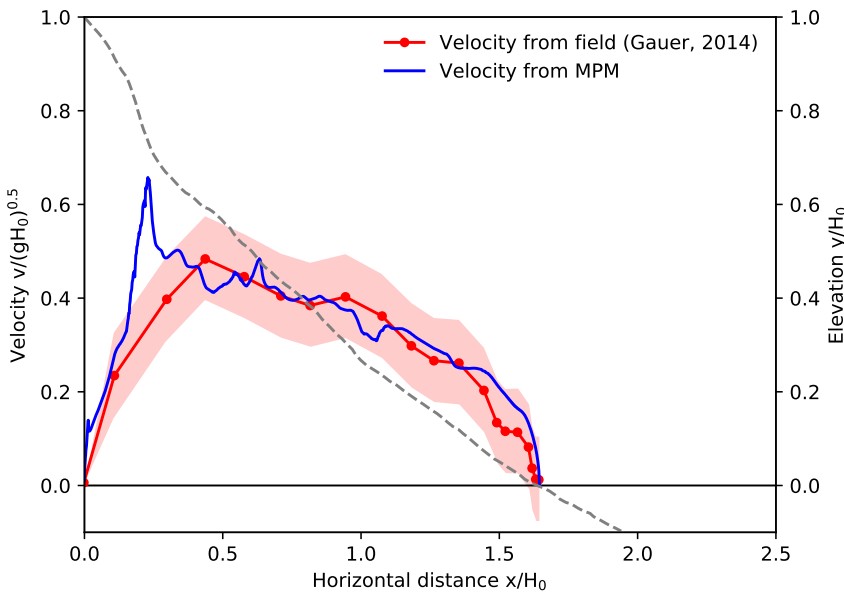

**Figure 14.** Front velocity distribution along the flow path for Case IV: Ryggfonn 2006-05-02 (Stryn, Norway). Drop height $H_0 = 303$ m.

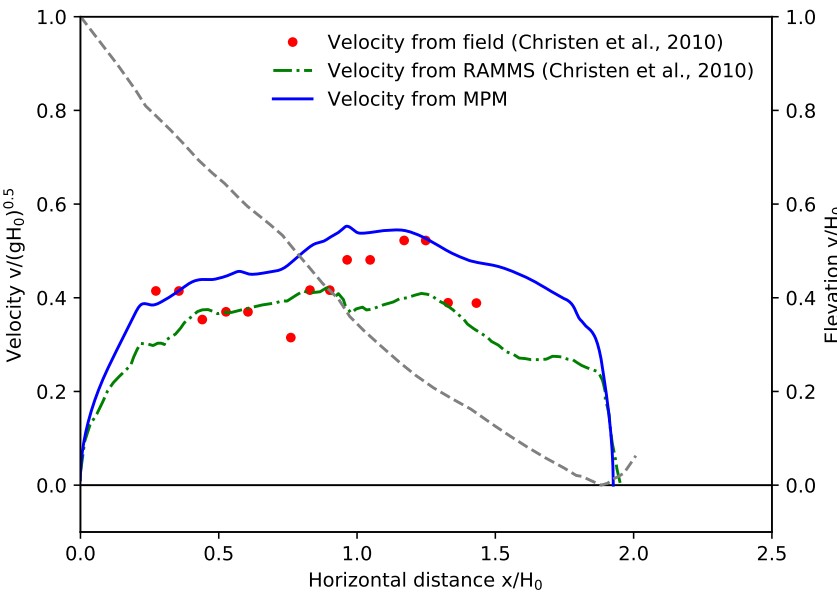

**Figure 15.** Front velocity distribution along the flow path for Case V: VdlS 2003-01-31 (Sion, Switzerland). Drop height $H_0 = 1246$ m.

Case IV in Fig. 14 is a snow avalanche composed of snow cornice at the release position and settled old snow in the track (Gauer, 2014). The consistency between the MPM data and the estimated velocity from a series of timed photographs (Gauer,

2014) is satisfactory, except for the overestimated velocity at the beginning of the flow. The overestimated front velocity from MPM is tightly related to the abruptly steepened slope at $x/H_0 \approx 0.2$. The increase of front velocity in reality was not as sharp as the MPM result, which might be due to the effect of more entrainment especially at the steep part of the slope. Indeed, it was reported that the maximum velocity of a simulated snow avalanche without entrainment is higher than that with entrainment, given the same runout distance (Sovilla and Bartelt, 2002; Sovilla et al., 2007). Moreover, the measurement data are based on photo series with intervals of 1 s, whilst the time gaps of the MPM data are 1/24 s. The maximum front velocity in reality might be lost within the 1 s interval of the measurement.

Fig. 15 demonstrates the data of Case V, including that from real measurement and RAMMS simulation (Christen et al., 2010) as well as our MPM modeling. The front velocity from the field was calculated from timed photographs. The dry snow avalanche in Case V was artificially triggered, and the development of powder part was reported (Christen et al., 2010). To be consistent, the adopted properties for the dry snow in Cases I and III are used for Case V. As shown in Fig. 15, both the RAMMS and MPM results show reasonable consistency with the real-measurement data. The calculated final front positions from RAMMS and MPM are similar, whilst the maximum front velocity is underestimated and overestimated by RAMMS and MPM, respectively. As discussed in Case IV, the MPM modeling does not take entrainment into account, which might be the reason of the overestimated front velocity.

## 5 Conclusions and discussion

This study explores the dynamics of snow avalanches with the Material Point Method (MPM) and an elastoplastic constitutive model. By virtue of the capability of the MPM in simulating processes with large deformations, fractures, collisions and coexistence of solid- and fluid-like behaviors, a wide range of distinct snow avalanches with diverse flow behaviors has been investigated. The reported four flow regimes for dense snow avalanches from real observations have all been captured from our MPM simulations, including cold shear, warm shear, warm plug and slab sliding regimes. Moreover, in transition from cold shear to warm shear flow regimes, flows with surges and small granules are observed. The evolution of the avalanche front, the free surface shape, and the vertical velocity profile shows distinct characteristics for the different flow regimes, underpinning the identification of flow regime. In addition to the flow surface and the shear behavior presented in this study, other features of the flow may also be used to pinpoint the flow regimes, such as snow temperature and liquid water content (Köhler et al., 2018). Although they are not explicitly taken into account in this study, the changing snow properties in our MPM modeling are capable of capturing the characteristics of the different regimes. Furthermore, distinct stopping mechanisms and maximum velocities were reported for the four regimes (Köhler et al., 2018). For example, the cold dense regime was identified by starving stopping mechanism, where the flow deposits and stops firstly from its tail then to its front. And the velocity of the cold dense regime was reported to be smaller than 30 m/s. It is noticed that the simulated flow with MPM does not fully follow these descriptions, which might be due to the idealized MPM setup and different terrain conditions.

We have systematically examined the effects of snow properties, slope angle, and path length on the flow and deposition behaviors of snow avalanches, including the maximum flow velocity on the slope, the runout angle and the avalanche deposit

height. It is found that snow friction and cohesion are closely related to the behavior of snow avalanches. Low snow friction and cohesion give fluid-like behavior and highly sheared flows, while high snow friction and cohesion lead to solid-like flow with limited shearing. Both slope angle and path length have a positive correlation with the maximum flow velocity on the slope, whilst their effects on the deposit height are trivial. Furthermore, unified trends have been obtained with normalization of the maximum flow velocity, the deposit height and the runout angle, revealing analogous physical rules under the different conditions. Key controlling factors of $v_{max}$ has been identified, including the friction between the bed and the flow, the geometry of the slope, as well as the snow properties. Depending on snow properties, the runout angle is either controlled by the flow behavior of a snow avalanche before its arrival at the deposition zone, or governed by its deposition behavior. It should be noted that a wide range of material parameters has been adopted for the sensitivity study. The combination of extreme flow properties leading to very high velocity might not be realistic for snow avalanches. The material parameters need to be carefully calibrated for investigation of real snow avalanches.

The MPM modeling has been calibrated and tested through simulations of real snow avalanches on irregular terrains. The calculated avalanche front position and velocity from MPM show reasonable agreement with the measurement data from literature. The behavior of dense snow avalanches has been well recovered by MPM. Discrepancy was observed particularly for avalanches which developed a powder cloud above the dense core, as the powder cloud has not been modeled here.

The presented research focuses on examining the flow regimes and flow dynamics of snow avalanches with idealized conditions, which is a preliminary study serving as the basis for investigating more realistic and complex snow avalanches. The 2D ideal slope with a constant inclination could be further changed to other shapes to be more realistic, such as parabolic track. Although the 2D setups were used to efficiently conduct the systematic study including more than 1000 cases, it is fully possible to explore interesting cases with 3D MPM simulations (Gaume et al., 2019). Future studies will take into account real topography in 3D and recover the natural boundary conditions of snow avalanches. In addition, a new framework will need to be developed for investigating snow avalanches with a powder cloud, by considering a new constitutive law for the cloud and its interaction with both the dense core of snow avalanches and the air around the cloud (e.g. air friction). To further consider entrainment, the snow cover could be explicitly simulated with our model. This would however significantly increase the computational time. Alternatively, one could add a mass flux rate term to the mass balance equation, which considers the snow cover as a rigid boundary and estimates the entrained mass based on empirical and theoretical relations (Naaim et al., 2004; Issler and Pérez, 2011). Meanwhile, the momentum conservation needs to be adjusted to account for the momentum change of snow avalanches due to entrainment. Despite the assumptions and idealization applied in this study, it is demonstrated that the MPM model provides a promising pathway towards systematic and quantitative investigations on snow avalanche dynamics and flow regime transitions under the effects of snow mechanical properties and terrain geometries, which can improve our understanding of wet snow avalanches and offer analysis for avalanche dynamics with the influence of climate change.

## Appendix A: Energy evolution of the flows in the four typical flow regimes

The constitutive model adopted in this study perfectly satisfies the second law of thermodynamics. Following the derivation in Gaume et al. (2018a), proving energy does not increase is equivalent to proving the plastic dissipation rate $\dot{w}^P(\boldsymbol{X}, t)$ is non-negative. $\dot{w}^P$ can be computed as

$$\dot{w}^P = -\boldsymbol{\tau} : \frac{1}{2}(\mathcal{L}_v \boldsymbol{b}^E)(\boldsymbol{b}^E)^{-1} \tag{A1}$$

where $\boldsymbol{\tau}$ is the Kirchhoff stress tensor, $\mathcal{L}_v$ is the Lie derivative, and $\boldsymbol{b}^E$ is the elastic right Cauchy-Green strain tensor. Since we use an associative flow rule, $\mathcal{L}_v \boldsymbol{b}^E = -2\dot{\gamma}\frac{\partial y}{\partial \boldsymbol{\tau}}\boldsymbol{b}^E$ (see Equation 10 in Gaume et al. (2018a)), $\dot{w}^P$ can be expressed as

$$\dot{w}^P = \boldsymbol{\tau} : \dot{\gamma}\frac{\partial y}{\partial \boldsymbol{\tau}} = \dot{\gamma}\hat{\boldsymbol{\tau}} \cdot \frac{\partial y}{\partial \hat{\boldsymbol{\tau}}} \tag{A2}$$

Recall Equation 11 in Gaume et al. (2018a) that $\dot{\gamma} \geq 0$. Furthermore, $\hat{\boldsymbol{\tau}} \cdot \frac{\partial y}{\partial \hat{\boldsymbol{\tau}}} \geq 0$ because the yield surface is a convex function of $\hat{\boldsymbol{\tau}}$ which includes the origin. Therefore $\dot{w}^P \geq 0$. Notes this result holds for any isotropic plasticity model that has a convex yield function and an associative flow rule.

The evolution of kinetic and potential energy of the flows in the four typical flow regimes (i.e. cold dense, warm shear, warm plug, sliding slab) is demonstrated in Fig. A1. As expected, the potential energy of the flows initially decreases as the flows move down from the slope, and then becomes steady after the flows stop. The kinetic energy of the flows firstly increases and then reduces until it vanishes. It is noticed that the kinetic energy of the sliding slab shows fluctuations in the deceleration phase, due to the interactions between the separating slabs in the flow after they reach the connecting arc zone (see supplementary video 1).

Fig. A2 shows the dissipated energy of the flows in the four cases. The dissipated energy increases before it reaches the final steady state. The growth rate of the dissipated energy varies for the different flows as they have distinct flow behaviors. Nevertheless, the final energy dissipation does not show much difference for the different flows. This is because of the identical initial potential energy and the similar final potential energy of the flows.

The energy dissipation is contributed from 1) internal force of the material and 2) external force on the material from the boundary/slope. As illustrated in Fig. A3, in all the four cases, the dissipated energy from the boundary is much higher than that dissipated inside the material. This is consistent with the results in Gracia et al. (2019).

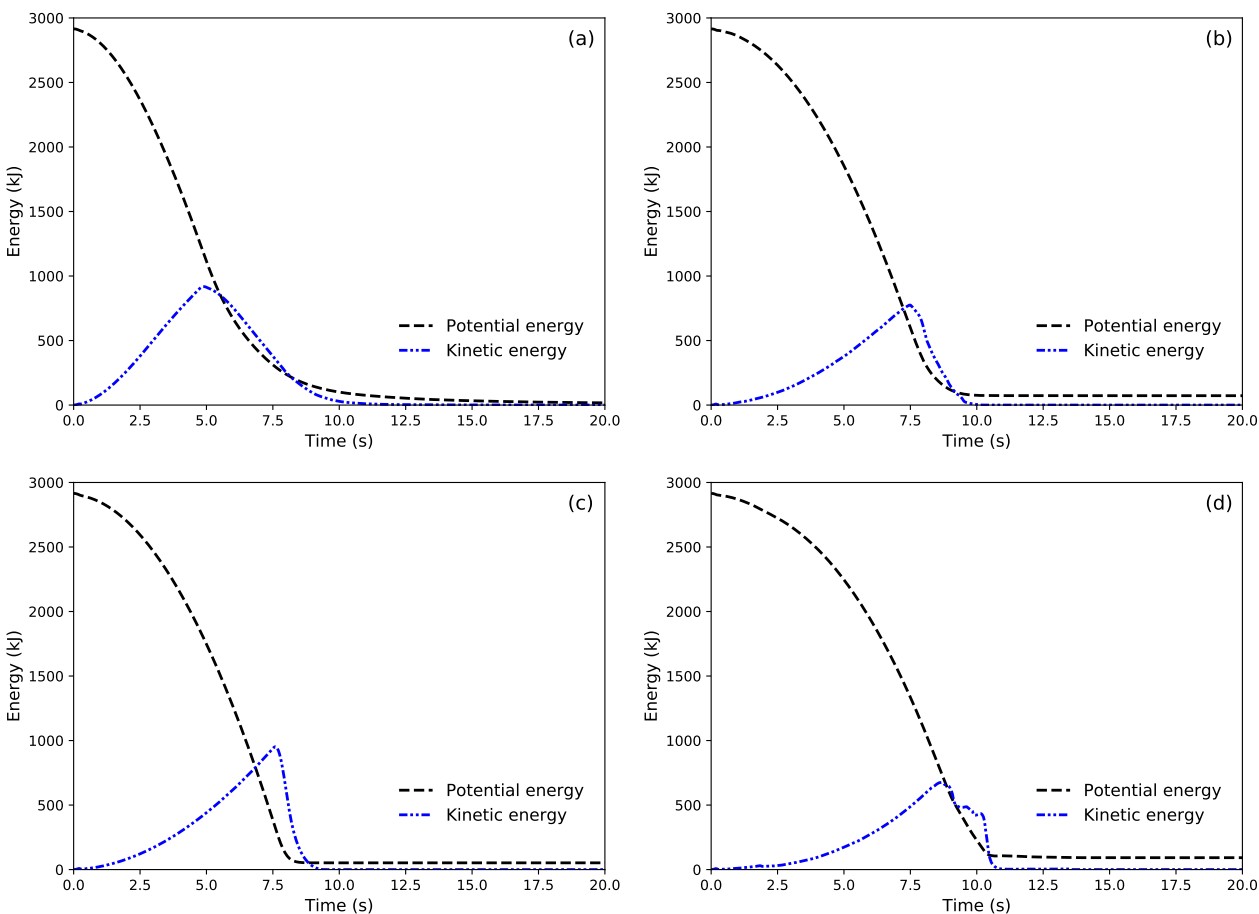

**Figure A1.** Evolution of potential and kinetic energy of the flows in the four typical flow regimes. (a) cold dense; (b) warm shear; (c) warm plug; (d) sliding slab.

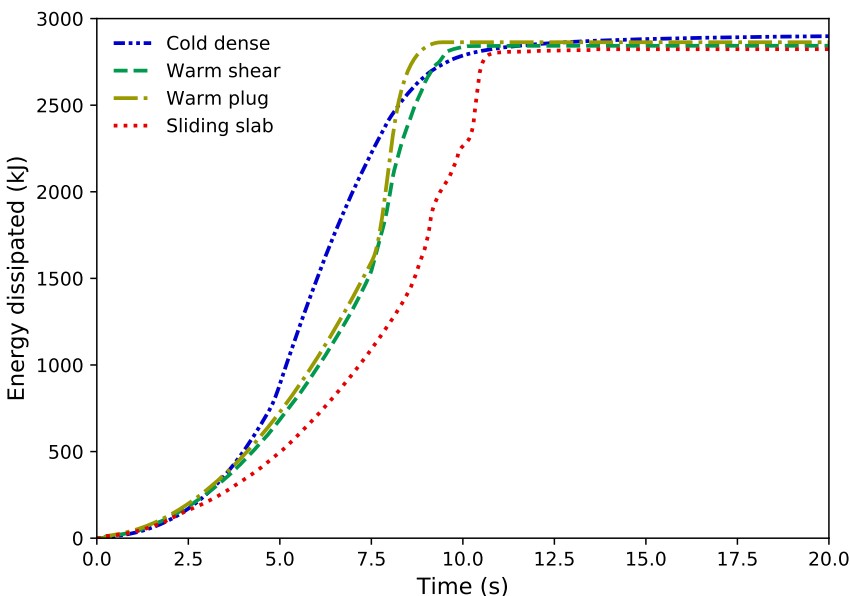

**Figure A2.** Evolution of dissipated energy of the flows in the four typical flow regimes.

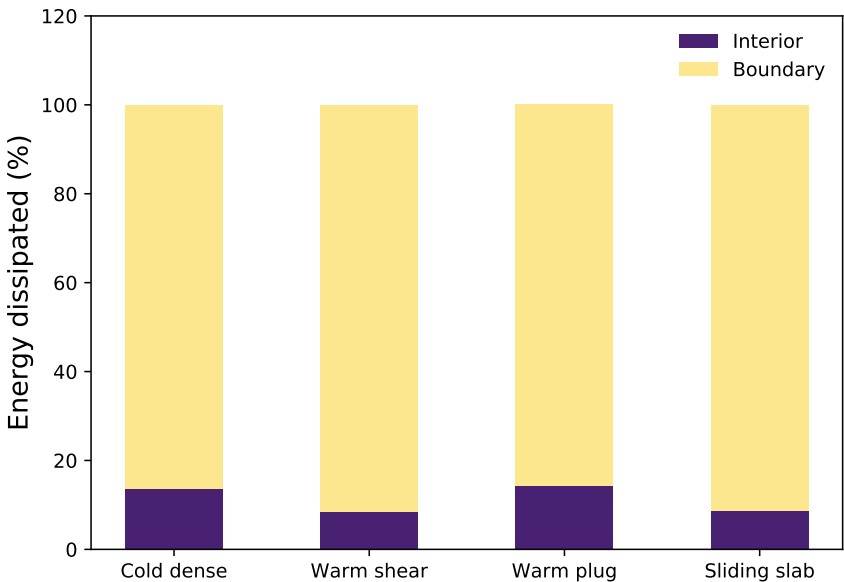

**Figure A3.** Energy dissipation inside the flow and through the boundary bed in the flows with the different flow regimes.

*Data availability.*  All the relevant data are available on Zenodo at https://doi.org/10.5281/zenodo.3965795.

*Video supplement.*  Movies of the avalanches presented in Fig. 2 are available on Zenodo at https://doi.org/10.5281/zenodo.3944698.

*Author contributions.*  J.G. designed the study and obtained the funding. X.L. conducted the study, performed the simulations and wrote the paper under the supervision of J.G.. B.S. commented on the paper and provided guidance on the flow regime transitions. C.J. developed the MPM tool, which has been used in this study.

*Competing interests.*  The authors declare that they have no conflict of interests.

*Acknowledgements.*  J.G. acknowledges financial support from the Swiss National Science Foundation (grant number PCEFP2_181227). The first author acknowledges Lars Blatny for his support on the language editing of this paper.

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
