# Peer review of "The mechanical origin of snow avalanche dynamics and flow regime transitions"

_The Cryosphere, 2020_

## Short Comment (SC1) · 28 May 2020

The paper by X. Li, B. Sovilla, C. Jiang and J. Gaume entitled The mechanical origin of snow avalanche dynamics and flow regime transitions is well organised and written with a short introduction, three sections presenting the different steps of the work and some conclusions and perspectives to finish with.

In the introduction, the applicative context is first depicted regarding the necessity of investigating the snow avalanche dynamics for a better understanding and protection of people and human goods. The originality of this study is justified by the need of having a numerical tool to model the dynamics of snow avalanches with snow of different types and different slope geometries.

[Figure]

In section 2, the MPM is briefly described, as well as the constitutive model mainly referring to former contributions by some of the authors but not solely.

Section 3 presents a complete parametric study of five snow types flowing along an ideal slopes and arresting on an horizontal plane. The inclination and length of the slope are also part of the parametric study. All simulations falls into four typical snow avalanche groups denoted cold dense, warm shear, warm plug and sliding slab. The front velocity, the velocity profile across the flow, the arresting distance and the free surface shape are part of the output parameters analysed. The results are qualitatively in agreement with the physics and discussed as such. The influence of the snow type is systematically explained. Unfortunately, only macroscopic quantities (see above) as output are studied to distinguish flow types. I would suggest, as in Gracia et al. (2019) [F. Gracia, P. Villard, V. Richefeu (2019) Comparison of two numerical approaches (DEM and MPM) applied to unsteady flow, Computational Particle Mechanics, 6(4), pp. 591-609] which deals with the same topic applied to granular flows, in order to understand the internal physics of the flow that you extract, show and discuss some quantities such as energies (potential, kinetic, dissipated by friction or fracture) to understand their transfers during the flow and to provide an insight to understand which material parameters, including the basal friction coefficient, are the key ones. Some master curves or should I say master clouds are proposed with dimensionless parameters. Proposition of analytical solutions fitting the simulated results would be an interesting point for further uses towards a quantitative step.

In section 4, the model strategy is applied to real cases with field measurements. It should be more clearly stated in each case what are the parameters that are set a priori and the one used for the calibration process. I suggest to set some stars in table 3 to distinguish calibrated parameters. The results are impressive with a very good agreement in general with field measures. The discrepancies are explained by the fact that MPM cannot entrain further material during the flow, that the turbulence dynamics in powder cloud is not modelled in MPM (some perspectives are set along this line

although the frictional dissipation with air is not mentioned), that the measurement acquisition frequencies are not comparable between field and numerical data (in order to be more precise on this point, data could be presented with points instead of lines, for instance in Fig 14 where the velocity peak is much discussed.)

The conclusion summarises the main qualitative results. A very interesting discussion is proposed at the end for the future work towards real geometry in 3D (MPM tools already exist in 3D, thus it is mainly a matter of computational time), to introduce in the MPM tool a constitutive law dedicated to powder cloud and its interaction with the dense part (the air friction is not mentioned here).

Overall the contribution is very well written, clear and well organised. The results and analysis are well documented, except the few points mentioned in bold in this review which need to be addressed for the final version. The work is original and provides an interesting step towards the prediction of snow avalanche propagation conditions.

---

## Referee Comment (RC2) · Anonymous Referee #2 · 24 Jun 2020

**Review of 'The mechanical origin of snow avalanche dynamics and flow regime transitions' by Xingyue Li, Betty Sovilla, Chenfanfu Jiang, and Johan Gaume 2018 - tc-2020-83**

**General comments:**

The paper presents a novel application of the authors recently developed approaches, successfully combining experimental findings on the flow regime evolution in snow avalanches and respective modelling approaches. The authors reach the goal of showing the models ability to replicate different flow regimes (and the associated flow

characteristics, such as velocity, ...) by tuning the corresponding material parameters.

One point that could be enhanced in my eyes is the discussion of the role and connection between the numerical method/solver and the applied flow/material model. As the title states, the paper aims at the identification of the *mechanical* rather then the numerical origin of flow regimes in snow avalanches. However, the numerical method/solver (MPM) is often highlighted and associated with the success of the modeling results rather than the corresponding material model (see comments below).

Overall the paper is very well written and includes helpful figures with corresponding supplementary material (with some small exceptions mentioned below). This valuable contribution is of high quality, enjoyable to read and fits to the scope of TC.

**Specific comments:**

- *p2 l 41-51 and section 2.1*: could you include a comment what the main differences (e.g. 2d/3d, depth resolved/averaged, ...) are to the classical, numerical approaches that are used in common simulation software that you also cite throughout your paper (such as Christen et al. (2010)). In particular the similarities and/or differences are to other particle based methods such as SPH (which are also used for classical shallow water 2d avalanche modelling Sampl and Zwinger (2004)) would probably be interesting for the reader to also interpret the future potential of the MPM methods (see conclusions).

- *p5 line 106, Table 1*: here you particularly highlight the parameters for the MPM modeling. To me it appears that this could be misleading. All parameters refer to the material model (section 2.2.). No numerical parameters are discussed - therefore the it would be interesting to: 1) comment the role of the numerical

parameters and how they where chosen and to 2) clarify the role/interplay of the numerical technique and the material model (see comment on paper title above).

- *p7 line 145*: Could you briefly explain a bit more what this threshold means - and if or if not this is connected to the (numerical?) fluctuations that appear e.g. in Figure 3 b) around 5s for the cold dense and 7.5-10s for the warm shear simulations?

- *p15 line 276*: Could you briefly comment on what the plateau stage means and if or if not any of the avalanches reach some kind of final velocity / steady state?

- *p16 l 291, ...To calibrate and benchmark our MPM modeling...*: is this really a calibration or rather a parameter variation/test with respect to the material / flow model rather than the numerical MPM approach?

- *p16 l 307-310*: I think here you have to clarify in more detail: 1) how are the avalanche velocities measures (different measurement techniques will lead to different velocities (front / core), see e.g. Rammer et al. (2007); Gauer et al. (2007)) and 2) if the measurements are comparable are the simulated velocities transformed correspondingly such they can be directly compared to the measurements (see e.g. Fischer et al. (2014))?

**Technical corrections:**

Generally text and Figures are clear and the supplementary material is very helpful. Possible corrections include:

- *Figure 2 and supplementary material*: Fig 2 is missing a spatial scale and the corresponding video is missing a legend (velocity/epsilon scale) as well as a spatial and temporal scale

- *Figures 11-15 and supplementary material*: absolute scales are missing and prohibit valuable data interpretation (at least total fall height should be stated in a Table or the caption)

- *wording*: $\alpha$ should be referred to as runout angle

- *wording*: H/L and $H_0/L_0$ should be referred to the other way around (H/L=tan $\alpha$ is usually the convention why H/L refers to the topography inclination in this paper)

- *wording*: what the authors refer to as "benchmark" appears more as a model "test" to me

- *wording*: please check by a native speaker if the choice of plural/singular is appropriate throughout the paper (e.g. behaviours, literatures, terrains, ...)

**References**

Christen, M., Kowalski, J., and Bartelt, P. (2010). RAMMS: Numerical simulation of dense snow avalanches in three-dimensional terrain. *Cold Regions Science and Technology*, 63:1–14.

Fischer, J. T., Fromm, R., Gauer, P., and Sovilla, B. (2014). Evaluation of probabilistic snow avalanche simulation ensembles with Doppler radar observations. *Cold Regions Science and Technology*, 97(0):151–158.

Gauer, P., Kern, M., Kristensen, K., Lied, K., Rammer, L., and Schreiber, H. (2007). On pulsed Doppler radar measurements of avalanches and their implication to avalanche dynamics. *Cold Regions Science and Technology*, 50(1):55–71.

Rammer, L., Kern, M., Gruber, U., and Tiefenbacher, F. (2007). Comparison of avalanche-velocity measurements by means of pulsed Doppler radar, continuous wave radar and optical methods. *Cold Regions Science and Technology*, 50(1-3):35–54.

Sampl, P. and Zwinger, T. (2004). Avalanche simulation with SAMOS. *Annals of Glaciology*, 38(1):393–398.

---

## Referee Comment (RC3) · Anonymous Referee #3 · 1 Jul 2020

The paper presents a systematic approach to evaluate the potential of the Material Point Method (MPM) for snow avalanches. The MPM method provides the possibility to account for different flow regimes of the avalanche flow in rather a novel approach.

In a first step the paper concerns avalanche on a selection of very simple geometries (At this point one could have considered a variety of parabolic track as this might be closer to Nature). The authors present a nice comparison of the influence of various parameters which would determine the flow regime.

I do have slight problems with section 3.3.2. First of all, the main idea behind so-called alpha-beta model (Lied and Bakkehoi, 1980) is that the runout angle alpha is proportional the the beta angle, which is a measure of the mean slope angle. Hence there is no dependency on a length scale in the runout. Furthermore, solely considering the

alpha angle involves little information without the corresponding beta angle. Having said that, Fig. 8 and Fig. 9 are not that easy to understand. For example, even though the velocities Fig. 9 seem to correspond somehow with the measurements, their origin (frictional behavior) might be rather different. E.g. the velocity of 70 m/s in the simulations correspond to nearly free fall velocity $(2gH)^{.5}$ whereas as the measured one is close to $(gH/2)^{.5}$. Hence there is a mix up in the comparison.

Finally, the authors present a promising comparison between simulations and real avalanche measurements.

It would be interesting to see how the model would behave when erosion and entertainment is also considered.

Some minor remarks can be found in the attachments.

Best regards

Please also note the supplement to this comment:
https://tc.copernicus.org/preprints/tc-2020-83/tc-2020-83-RC3-supplement.pdf

**Supplement:**

[revised manuscript text omitted]

---

## Author Comment (AC3) · 16 Jul 2020

**Response to Reviewer #3's comment on "The mechanical origin of snow avalanche dynamics and flow regime transitions"**

Xingyue Li, Betty Sovilla, Chenfanfu Jiang, and Johan Gaume*
*Correspondence: johan.gaume@epfl.ch
July 15th, 2020

We thank Referee #3 for his or her detailed comments and valuable suggestions, which helped us to improve the quality of the paper. Our point-to-point replies to the comments of the reviewer are summarized below.

*The paper presents a systematic approach to evaluate the potential of the Material Point Method (MPM) for snow avalanches. The MPM method provides the possibility to account for different flow regimes of the avalanche flow in rather a novel approach.*

*In a first step the paper concerns avalanche on a selection of very simple geometries (At this point one could have considered a variety of parabolic track as this might be closer to Nature). The authors present a nice comparison of the influence of various parameters which would determine the flow regime.*

Reply: We thank this reviewer for the suggestion on the parabolic track, which could be interesting and more realistic for future studies on snow avalanches with MPM modelling. This has been added to the discussion.

*I do have slight problems with section 3.3.2. First of all, the main idea behind so-called alpha-beta model (Lied and Bakkehoi, 1980) is that the runout angle alpha is proportional the beta angle, which is a measure of the mean slope angle. Hence there is no dependency on a length scale in the runout. Furthermore, solely considering the alpha angle involves little information without the corresponding beta angle. Having said that, Fig. 8 and Fig. 9 are not that easy to understand. For example, even though the velocities Fig. 9 seem to correspond somehow with the measurements, their origin (frictional behavior) might be rather different. E.g. the velocity of 70 m/s in the simulations correspond to nearly free fall velocity (2gH)ˆ.5 whereas as the measured one is close to (gH/2)ˆ.5. Hence there is a mix up in the comparison.*

Reply: The following offers our point-to-point response to the comments on section 3.3.2.

1) Dependency of $\alpha$ on length scale: We agree that the runout angle $\alpha$ highly depends on the mean slope angle $\beta$. According to Lied and Bakkehøi (1980), the following correlation between $\alpha$ and $\beta$ was obtained based on 111 avalanches

$$\alpha = 0.97\beta - 1.4°$$

with a standard deviation of 3.5° and R = 0.88. However, a more accurate prediction of α was reported as follows in Lied and Bakkehøi (1980)

$$\alpha = (6.2 \times 10^{-1} - 2.8 \times 10^{-1}Hy'')\beta + (1.9 \times 10^{1}Hy'' - 2.3)° + 1.2 \times 10^{-1}\theta$$

which has a standard deviation of 2.3° and R = 0.95. H is the total vertical displacement. y'' is the terrain profile of the avalanche path described by the second derivative. θ is the inclination of the starting zone (Note the θ in Lied and Bakkehøi (1980) has a different definition with the slope angle θ in our study). Thus, the runout angle also depends on the length scale of the avalanche path in addition to the mean slope. As stated in Lied and Bakkehøi (1980), "*The most important parameter is the β. Hy'' is also an important parameter*".

2) Discussion of α without mentioning β: As reviewed above, the average slope angle β is a very important factor controlling the runout angle α. The origin/reason for proposing β is to describe the mean slope angle of a complicated and irregular flow path which is normally the case in reality. In our study, ideal slopes are used for the sensitivity study, whose mean slope angle β is very close to their actual slope angle (θ in our manuscript). We initially discussed the effect of θ without mentioning β to avoid the repetition. The relation between β and θ has been clarified in the revised manuscript. In addition, it is found that the positive correlation between the maximum runout angle and the slope angle from MPM in Fig. 8 agrees with the α-β model, which has been mentioned in the revised manuscript.

3) Comparison of flow velocities from MPM and real measurements: As mentioned in Lines 269-270 in the manuscript, the real avalanche with a velocity of 70 m/s was a powder snow avalanche, whose dense core can be captured by the current MPM model while the powder cloud is beyond the scope of this study. We agree that the high velocities (close to 70 m/s) from the real avalanche and the simulated avalanche come from different physical processes. The high velocity of the real avalanche is resulted from the large drop height (1940 m from McClung and Gauer (2018)). In contrast, the high velocity of the simulated avalanche is mainly controlled by the properties (low friction and low cohesion) of the flow. While we observe a generally fair agreement of the MPM and field data in Fig. 9, a quantitative comparison would require full consistency of the model setup (e.g. drop height, flow properties), as we did in Section 4 of the paper. Our main motivation here is to show the influence of mechanical (M and β) and geometrical (θ and $L_0$) properties on the $v_{max}$-α relationship and give a new insight to the negative correlation observed from the data in McClung and Gauer (2018) (Lines 273-277).

According to the relation between the flow velocity and the drop height reported for real snow avalanches (Gauer, 2014), the high flow velocity close to 70 m/s obtained with a drop height of 211.2 m from the MPM simulation might not be realistic for snow avalanches. It has been clarified in the revision that the adopted material parameters are designed to study a wide range of different material properties, while

the cases with very low friction M and cohesion β leading to the very high velocity might not be realistic for snow avalanches. The material parameters need to be carefully calibrated for investigation of real snow avalanches.

References:
- Gauer, P. (2014). Comparison of avalanche front velocity measurements and implications for avalanche models. *Cold Regions Science and Technology*, 97:132-150.
- Lied, K., and Bakkehøi, K. (1980). Empirical calculations of snow–avalanche run–out distance based on topographic parameters. *Journal of Glaciology*, 26(94):165-177.
- McClung, D. M., and Gauer, P. (2018). Maximum frontal speeds, alpha angles and deposit volumes of flowing snow avalanches. *Cold Regions Science and Technology*, 153:78-85.

*Finally, the authors present a promising comparison between simulations and real avalanche measurements.*

*It would be interesting to see how the model would behave when erosion and entertainment is also considered.*

*Some minor remarks can be found in the attachments.*

Reply: We appreciate the reviewer's interest in the performance of the model with consideration of erosion and entrainment. We also consider entrainment as an interesting and very important process in snow avalanches, and will be the topic of our next study using MPM (Lines 371-375).

*Specific comments:*

*1. Line 9: "Each of the flow regimes shows" should be "Each of the flow regimes show"?*

Reply: Since "each" is our subject, "shows" is used.

*2. Line 13: "scaled α angle", an angle can hardly be scaled.*

Reply: The scaled α angle refers to the dimensionless α*. This notation has been clarified in the revision.

*3. Line 14: "It is found …" to "It is found that …".*

Reply: Revised.

*4. Line 29: "classified" to "considered".*

Reply: Revised.

*5. Line 30: Delete "including".*

Reply: "including" has been replaced by "namely,".

6. Line 30: "Recent study" to "A recent study".

Reply: Revised.

7. Line 36: "tools", model (I think is the better word here).

Reply: Thanks for the rewording. Revised.

8. Fig. 2 caption: Add "(Tab. 1, Group II)".

Reply: Added.

9. Line 146: "front position" to "the front position".

Reply: Revised.

10. Fig. 3 caption: Add "(Tab. 1, Group II)".

Reply: Added.

11. Fig. 3: Which mu value is used in the calculation of $v_{max}^b$?

Reply: Thanks for the question. The value of mu is 0.5, which has been added to the text and to Table 1.

12. Fig. 4 caption: Add "(Tab. 1, Group II)".

Reply: Added.

13. Fig. 5 caption: Add "and lengths L".

Reply: The data with different lengths L ($L_0$ in the revised manuscript) are not included in Fig. 5. They are plotted in Fig. 6.

14. Fig. 6 caption: Add "slope angles".

Reply: The data with different slope angles are not included in Fig. 6. They are plotted in Fig. 5.

15. Line 226: "which hints an analogous physical rule behind the trend", what is the physical rule you are thinking of?

Reply: As described in Lines 225-246, the data from the different groups of simulations give a similar trend, which drives us to normalize the results and find the analogous physical rule behind the similarity. Based on the normalized results in Fig. 7, there are different physical processes governing the data in the different regions. The maximum velocity of the cases close to the zero line in Fig. 7 is controlled by the friction between the flow and the bed (Lines 238-239). On the other hand, the velocity of the cases with small M and β in Fig. 7 is governed by the snow properties (Lines 239-244). Furthermore, the velocity of the cases far below the zero line in Fig. 7 is due to an acceleration smaller than the theoretical one obtained from a block sliding over a frictional bed (Lines 244-246). All the cases from the different groups follow and share these three physical processes.

*16. Figs 8&9: I'm wondering how the graphs would look like for H = L*tan(slope) combined.*

Reply: We tried to plot all the data in one figure for the varying H ($H_0$ in the revised manuscript and hereafter) as shown in Fig. 1 below. The increase of drop height does not necessarily give an increasing maximum velocity if we compare the data with $H_0$ = 73.5 m and the data with $H_0$ = 132.0 m. This is because these two groups do not have the same slope angle in this study. Thus, it is necessary to separately discuss the groups with a fixed slope angle and the groups with a fixed horizontal length, as we did in Figs 8&9. It is mentioned in the revised manuscript that, instead of fixing the horizontal length L ($L_0$ in the revised manuscript) when the slope angle is changed (Groups I, II, III in Table I), one could fix the vertical drop height $H_0$ and change the horizontal length.

[Figure]

Figure 1. Evolution of the maximum velocity with α for varying drop height $H_0$.

*17. Line 355: "Both slope angle and path length have a positive correlation with the maximum front velocity on the slope", this is not that surprise as total Drop height  H = L\*tan(slope).  and Umax \prop f(H).*

Reply: We agree that the total drop height should have a similar effect as the horizontal length L and the slope angle θ. The reason that we separately discuss the slope angle and the path length is that we have both the slope angle and the path length in the calculation of the theoretical maximum velocities $v_{max}^b$ and $v_{max}^f$ (Lines 152-156).

---

## Author Response (AR1)

**Response to Editor Florent Domine's comment on "The mechanical origin of snow avalanche dynamics and flow regime transitions"**

Xingyue Li, Betty Sovilla, Chenfanfu Jiang, and Johan Gaume*
*Correspondence: johan.gaume@epfl.ch
August 4th, 2020

We thank the editor for his positive evaluation of our responses and his suggestion on the manuscript. Our detailed reply is provided below. The point-to-point responses to the reviewers' comments have been attached as we have added line numbers to locate our answers. In addition, a marked-up manuscript with tracked changes is attached.

*Editor:*

*Thank you for your detailed and adequate responses to the Reviewers' comments. Please upload your revised version. Please also be careful to read our instructions regarding the difference between supplementary material and appendices.*

*https://www.the-cryosphere.net/for_authors/manuscript_preparation.html*

*It is likely that what you plan to add as supplementary material in your response to Reviewer 1 should in fact be included as an appendix.*

*I look forward to reading your revised version.*

Reply: We have added the planned supplement about energy evolution as an appendix in the revised manuscript (Pages 24-26). Correspondingly, we have cited the appendix in Lines 156-157 in the revision.

**Response to Reviewer #1 Frederic Dufour's comment**

We want to thank Prof. Dufour for his comments and his constructive suggestions that helped us to improve the quality of our paper. In the following, we provide detailed point-by-point answers to the comments raised by the reviewer.

*The paper by X. Li, B. Sovilla, C. Jiang and J. Gaume entitled The mechanical origin of snow avalanche dynamics and flow regime transitions is well organised and written with a short introduction, three sections presenting the different steps of the work and some conclusions and perspectives to finish with.*

Reply: We thank the reviewer for this positive evaluation of our paper.

*In the introduction, the applicative context is first depicted regarding the necessity of investigating the snow avalanche dynamics for a better understanding and protection of people and human goods. The originality of this study is justified by the need of having a numerical tool to model the dynamics of snow avalanches with snow of different types and different slope geometries.*

*In section 2, the MPM is briefly described, as well as the constitutive model mainly referring to former contributions by some of the authors but not solely.*

*Section 3 presents a complete parametric study of five snow types flowing along ideal slopes and arresting on a horizontal plane. The inclination and length of the slope are also part of the parametric study. All simulations fall into four typical snow avalanche groups denoted cold dense, warm shear, warm plug and sliding slab. The front velocity, the velocity profile across the flow, the arresting distance and the free surface shape are part of the output parameters analysed. The results are qualitatively in agreement with the physics and discussed as such. The influence of the snow type is systematically explained. Unfortunately, only macroscopic quantities (see above) as output are studied to distinguish flow types. I would suggest, as in Gracia et al. (2019) [F. Gracia, P. Villard, V. Richefeu (2019) Comparison of two numerical approaches (DEM and MPM) applied to unsteady flow, Computational Particle Mechanics, 6(4), pp. 591-609] which deals with the same topic applied to granular flows, in order to understand the internal physics of the flow that you extract, show and discuss some quantities such as energies (potential, kinetic, dissipated by friction or fracture) to understand their transfers during the flow and to provide an insight to understand which material parameters, including the basal friction coefficient, are the key ones. Some master curves or should I say master clouds are proposed with dimensionless parameters. Proposition of analytical solutions fitting the simulated results would be an interesting point for further uses towards a quantitative step.*

Reply: The basis for analysing energy is energy conservation. The constitutive model adopted in this study perfectly satisfies the second law of thermodynamics (Line 119-120 in the marked-up manuscript). Following the derivation in Gaume et al. (2018),

proving that energy does not increase is equivalent to proving the plastic dissipation rate $\dot{w}^P(\boldsymbol{X}, t)$ is non-negative. $\dot{w}^P$ can be computed as

$$\dot{w}^P = -\boldsymbol{\tau} : \frac{1}{2}(\mathcal{L}_v \boldsymbol{b}^E)(\boldsymbol{b}^E)^{-1}$$

where $\boldsymbol{\tau}$ is the Kirchhoff stress tensor, $\mathcal{L}_v$ is the Lie derivative, and $\boldsymbol{b}^E$ is the elastic right Cauchy-Green strain tensor. Since we use an associative flow rule, $\mathcal{L}_v \boldsymbol{b}^E = -2\dot{\gamma}\frac{\partial y}{\partial \boldsymbol{\tau}}\boldsymbol{b}^E$ (see Equation 10 in Gaume et al. (2018)), $\dot{w}^P$ can be expressed as

$$\dot{w}^P = \boldsymbol{\tau} : \dot{\gamma}\frac{\partial y}{\partial \boldsymbol{\tau}} = \dot{\gamma}\hat{\boldsymbol{\tau}} \cdot \frac{\partial y}{\partial \hat{\boldsymbol{\tau}}}$$

Recall that $\dot{\gamma} \geq 0$ in Equation 11 in Gaume et al. (2018). Furthermore, $\hat{\boldsymbol{\tau}} \cdot \frac{\partial y}{\partial \hat{\boldsymbol{\tau}}} \geq 0$ because our yield surface is a convex function of $\hat{\boldsymbol{\tau}}$ which includes the origin. Therefore $\dot{w}^P \geq 0$. Note that this result holds for any isotropic plasticity model that has a convex yield function and an associative flow rule.

[Figure]

Figure 1. Evolution of potential and kinetic energy of the flows in the four typical flow regimes.

The evolution of kinetic and potential energy of the flows in the four typical flow regimes (i.e. cold dense, warm shear, warm plug, sliding slab) is shown in Fig. 1. As expected, the potential energy of the flows initially decreases as the flows move down from the slope, and then becomes steady after the flows stop. The kinetic energy of the flows

firstly increases and then reduces until it vanishes. It is noticed that the kinetic energy of the sliding slab shows fluctuations in the deceleration phase, due to the interactions between the separating slabs in the flow after they reach the connecting arc zone (see supplementary video 1).

Fig. 2 shows the dissipated energy of the flows in the four cases. The dissipated energy increases before it reaches the final steady state. The growth rate of the dissipated energy varies for the different flows as they have distinct flow behaviours. Nevertheless, the final energy dissipation does not show much difference for the different flows. This is because of the identical initial potential energy and the similar final potential energy of the flows.

[Figure]

Figure 2. Evolution of dissipated energy of the flows in the four typical flow regimes.

The energy dissipation is contributed from 1) internal force of the material and 2) external force on the material from the boundary/slope. As illustrated in Fig. 3, in all the four cases, the dissipated energy from the boundary is much higher than that dissipated inside the material. This is consistent with the results in Gracia et al. (2019).

From the above discussion, we can indeed get more information about the energies. However, we did not find contrasting distinction characterizing the different flow regimes of the flows. Therefore, we put the above discussion as an appendix (Page 24-26) cited in Lines 156-157.

Regarding the analytical solutions fitting the simulated results, we indeed thought about proposing analytical relations between the scaled maximum velocity and the scaled deposit height in Fig. 7 in our manuscript as well as between the scaled maximum velocity and the scaled runout angle in Fig. 10. However, the physical processes involved are strongly non-linear and too complicated to develop analytical solutions. For

example, the deposit height and the runout distance are greatly affected by multiple processes during the flow and deposition, including breakage and granulation of snow, surging, and piling up. While we propose highly simplified analytical solutions based on the block sliding theory, as a limit case, the development of a complete analytical model taking into account all previously mentioned processes is beyond the scope of this study.

[Figure]

Figure 3. Energy dissipation inside the flow and through the boundary bed in the flows with the different flow regimes.

References:

- Gaume, J., Gast, T., Teran, J., van Herwijnen, A., and Jiang, C. (2018). Dynamic anticrack propagation in snow. *Nature Communications*, 9(1):1-10.
- Gracia, F., Villard, P., and Richefeu, V. (2019). Comparison of two numerical approaches (DEM and MPM) applied to unsteady flow. *Computational Particle Mechanics*, 6(4):591-609.

*In section 4, the model strategy is applied to real cases with field measurements. It should be more clearly stated in each case what are the parameters that are set a priori and the one used for the calibration process. I suggest setting some stars in table 3 to distinguish calibrated parameters. The results are impressive with a very good agreement in general with field measures. The discrepancies are explained by the fact that MPM cannot entrain further material during the flow, that the turbulence dynamics in powder cloud is not modelled in MPM (some perspectives are set along this line although the frictional dissipation with air is not mentioned), that the measurement acquisition frequencies are not comparable between field and numerical data (in order to be more precise on this point, data could be presented with points instead of lines, for instance in Fig 14 where the velocity peak is much discussed.)*

Reply: Following the reviewer's suggestion, it has been clarified in Lines 350-351 in the revised manuscript that the bed friction of the slope is the only calibrated parameter. In addition, a star has been used in Table 3 to notify the calibrated parameter. The adopted snow properties are fixed according to the description of the snow type in the literature, as detailed at Lines 345-350.

We thank the reviewer for pointing out the importance of frictional dissipation with air in the discussion of powder cloud. The corresponding sentence has been modified to reflect this aspect (Line 439).

Scattered points connected with a line have been used to plot the real measurement data in Figs 11-14 in the revised manuscript, which indeed offer more information on the measurement acquisition frequencies. Please note we still use lines for the MPM simulation data, since adding points does not differ much from the pure lines because the points overlap with one another as shown in Fig. 4 (Fig. 14 in the manuscript) below.

[Figure]

Figure 4. Front velocity distribution along the flow path for Case IV: Ryggfonn 2006-05-02 (Stryn, Norway). Drop height $H_0$ = 303 m.

*The conclusion summarises the main qualitative results. A very interesting discussion is proposed at the end for the future work towards real geometry in 3D (MPM tools already exist in 3D, thus it is mainly a matter of computational time), to introduce in the MPM tool a constitutive law dedicated to powder cloud and its interaction with the dense part (the air friction is not mentioned here).*

Reply: The air friction has been added (Line 439).

*Overall the contribution is very well written, clear and well organised. The results and analysis are well documented, except the few points mentioned in bold in this review which need to be addressed for the final version. The work is original and provides an interesting step towards the prediction of snow avalanche propagation conditions.*

Reply: We thank the reviewer for his constructive comments that helped us to improve the quality of our paper.

**Response to Reviewer #2's comment**

We thank Referee #2 for his or her insightful comments and helpful advice, which increase the quality of our paper. The following provides our point-to-point responses to the general comments, specific comments, and technical corrections from the reviewer.

*General comments:*

*The paper presents a novel application of the authors recently developed approaches, successfully combining experimental findings on the flow regime evolution in snow avalanches and respective modelling approaches. The authors reach the goal of showing the models ability to replicate different flow regimes (and the associated flow characteristics, such as velocity, ...) by tuning the corresponding material parameters.*

*One point that could be enhanced in my eyes is the discussion of the role and connection between the numerical method/solver and the applied flow/material model. As the title states, the paper aims at the identification of the mechanical rather than the numerical origin of flow regimes in snow avalanches. However, the numerical method/solver (MPM) is often highlighted and associated with the success of the modeling results rather than the corresponding material model (see comments below).*

*Overall the paper is very well written and includes helpful figures with corresponding supplementary material (with some small exceptions mentioned below). This valuable contribution is of high quality, enjoyable to read and fits to the scope of TC.*

Reply: We thank this reviewer for the encouraging comments. Regarding the numerical framework and the material model, this study indeed focuses on the material model, as we mainly investigate the effect of material property in addition to slope geometry. The relation between the numerical framework and the material model has been clarified in the revised manuscript as detailed in the reply of specific comment 2 below.

*Specific comments:*

*1. p2 l 41-51 and section 2.1: could you include a comment what the main differences (e.g. 2d/3d, depth resolved/averaged, ...) are to the classical, numerical approaches that are used in common simulation software that you also cite throughout your paper (such*

*as Christen et al. (2010)). In particular the similarities and/or differences are to other particle based methods such as SPH (which are also used for classical shallow water 2d avalanche modelling Sampl and Zwinger (2004)) would probably be interesting for the reader to also interpret the future potential of the MPM methods (see conclusions).*

*References:*
*Christen, M., Kowalski, J., and Bartelt, P. (2010). RAMMS: Numerical simulation of dense snow avalanches in three-dimensional terrain. Cold Regions Science and Technology, 63:1–14.*
*Sampl, P. and Zwinger, T. (2004). Avalanche simulation with SAMOS. Annals of Glaciology, 38(1):393–398.*

Reply: We have provided further introduction of existing numerical approaches for snow avalanche modelling in the revised manuscript (Lines 43-59), including 2D, 3D, and particle-based continuum methods, as follows.

[revised manuscript text omitted]

*2. p5 line 106, Table 1: here you particularly highlight the parameters for the MPM modeling. To me it appears that this could be misleading. All parameters refer to the material model (section 2.2.). No numerical parameters are discussed therefore the it would be interesting to: 1) comment the role of the numerical parameters and how they were chosen and to 2) clarify the role/interplay of the numerical technique and the material model (see comment on paper title above).*

Reply: Indeed, the parameters in Table 1 include snow parameters. In addition to that, the information of slope geometry is also listed. Numerical parameters (i.e. mesh size, time step, and frame rate) have been added to Table 1 in the revision (Page 6). To avoid the confusion of "MPM model" and "material model", "Model parameters" in the title of Table 1 has been revised to "Parameters".

1) Numerical parameters govern the accuracy and stability of the modelling. The determination of the adopted numerical parameters (i.e. background mesh size, time step, and frame rate) has been detailed in Lines 138-140. The size of the background Eulerian mesh in MPM is selected to be small enough to guarantee the simulation accuracy and resolution, and meanwhile be large enough to shorten the computation time. The time step is constrained by the CFL condition and the elastic wave speed to secure the simulation stability. The simulation data are exported every 1/24 s.

2) The relation between the numerical framework and the material model has been clarified at Lines 93-98. Different material models can be implemented to the MPM numerical framework to simulate different processes. For example, a non-associated Mohr-Coulomb model was applied to model landslide and dam failure (Zabala and Alonso, 2011; Soga et al., 2016), and a non-associated Drucker-Prager model was used to simulate sand (Klár et al., 2016). In this study, we specifically use the associated Modified Cam Clay model developed for snow, which reproduces mixed-mode snow fracture and compaction hardening (Gaume et al., 2018). The important role of the material/constitutive model has also been clarified in Lines 67-68, 71.

[Figure]

Figure 5. Front velocity distribution along the flow path for Case I: Weissfluh-Northridge 1982-03-12 a1 (Davos, Switzerland). Drop height $H_0$ = 236 m.

[Figure]

Figure 6. Front velocity distribution along the flow path for Case II: Weissfluh-Northridge 1982-03-12 a2 (Davos, Switzerland). Drop height $H_0$ = 177 m.

[Figure]

Figure 7. Front velocity distribution along the flow path for Case III: Himmelegg 1990-02-14 (Alberg, Austria). Drop height $H_0$ = 352 m.

[Figure]

Figure 8. Front velocity distribution along the flow path for Case IV: Ryggfonn 2006-05-02 (Stryn, Norway). Drop height $H_0$ = 303 m.

[Figure]

Figure 9. Front velocity distribution along the flow path for Case V: VdlS 2003-01-31 (Sion, Switzerland). Drop height $H_0$ = 1246 m.

*Technical corrections:*

*Generally text and Figures are clear and the supplementary material is very helpful. Possible corrections include:*

*1. Figure2 and supplementary material: Fig2 is missing a spatial scale and the corresponding video is missing a legend (velocity/epsilon scale) as well as a spatial and temporal scale.*

Reply: Spatial scale has been added to Fig. 2. The supplementary videos have been revised to include spatial and temporal scale as well as legend.

*2. Figures 11-15 and supplementary material: absolute scales are missing and prohibit valuable data interpretation (at least total fall height should be stated in a Table or the caption).*

Reply: Drop height has been clarified in the caption of Figs 11-15 and added to the supplementary data.

*3. Wording: α should be referred to as runout angle.*

Reply: "α angle" has been revised to "runout angle".

*4. Wording: H/L and H0/L0 should be referred to the other way around (H/L=tan α is usually the convention why H/L refers to the topography inclination in this paper).*

Reply: The definitions of H/L and H0/L0 have been exchanged.

*5. Wording: what the authors refer to as "benchmark" appears more as a model "test" to me.*

Reply: "MPM model is benchmarked" in the abstract has been revised to "MPM modeling is calibrated and tested" (Line 18). "To calibrate and benchmark our MPM modeling" in Line 342 has been modified to "To testify the capability of the MPM modeling in capturing key dynamic features (i.e. front velocity and position) of snow avalanches" (Lines 342-343).

*6. Wording: please check by a native speaker if the choice of plural/singular is appropriate throughout the paper (e.g. behaviours, literatures, terrains, ...).*

Reply: Thanks for the reminder. We have checked and revised the words with a native speaker.

**Response to Reviewer #3's comment**

We thank Referee #3 for his or her detailed comments and valuable suggestions, which helped us to improve the quality of the paper. Our point-to-point replies to the comments of the reviewer are summarized below.

*The paper presents a systematic approach to evaluate the potential of the Material Point Method (MPM) for snow avalanches. The MPM method provides the possibility to account for different flow regimes of the avalanche flow in rather a novel approach.*

*In a first step the paper concerns avalanche on a selection of very simple geometries (At this point one could have considered a variety of parabolic track as this might be closer to Nature). The authors present a nice comparison of the influence of various parameters which would determine the flow regime.*

Reply: We thank this reviewer for the suggestion on the parabolic track, which could be interesting and more realistic for future studies on snow avalanches with MPM modelling. This has been added to Lines 434-435 in the discussion.

*I do have slight problems with section 3.3.2. First of all, the main idea behind so-called alpha-beta model (Lied and Bakkehoi, 1980) is that the runout angle alpha is proportional the beta angle, which is a measure of the mean slope angle. Hence there is no dependency on a length scale in the runout. Furthermore, solely considering the alpha angle involves little information without the corresponding beta angle. Having said that, Fig. 8 and Fig. 9 are not that easy to understand. For example, even though the velocities Fig. 9 seem to correspond somehow with the measurements, their origin (frictional behavior) might be rather different. E.g. the velocity of 70 m/s in the simulations correspond to nearly free fall velocity (2gH)ˆ.5 whereas as the measured one is close to (gH/2)ˆ.5. Hence there is a mix up in the comparison.*

Reply: The following offers our point-to-point response to the comments on section 3.3.2.

1) Dependency of α on length scale: We agree that the runout angle α highly depends on the mean slope angle β. According to Lied and Bakkehøi (1980), the following correlation between α and β was obtained based on 111 avalanches
$$\alpha = 0.97\beta - 1.4°$$
with a standard deviation of 3.5° and R = 0.88. However, a more accurate prediction of α was reported as follows in Lied and Bakkehøi (1980)
$$\alpha = (6.2 \times 10^{-1} - 2.8 \times 10^{-1}Hy'')\beta + (1.9 \times 10^{1}Hy'' - 2.3)° + 1.2 \times 10^{-1}\theta$$
which has a standard deviation of 2.3° and R = 0.95. H is the total vertical displacement. y'' is the terrain profile of the avalanche path described by the second derivative. θ is the inclination of the starting zone (Note the θ in Lied and Bakkehøi (1980) has a different definition with the slope angle θ in our study). Thus, the runout angle also depends on the length scale of the avalanche path in addition to the

mean slope. As stated in Lied and Bakkehøi (1980), "*The most important parameter is the β. Hy'' is also an important parameter*".

2) Discussion of α without mentioning β: As reviewed above, the average slope angle β is a very important factor controlling the runout angle α. The origin/reason for proposing β is to describe the mean slope angle of a complicated and irregular flow path which is normally the case in reality. In our study, ideal slopes are used for the sensitivity study, whose mean slope angle β is very close to their actual slope angle (θ in our manuscript). We initially discussed the effect of θ without mentioning β to avoid the repetition. The relation between β and θ has been clarified in Lines 311-313 in the revised manuscript. In addition, it is found that the positive correlation between the maximum runout angle and the slope angle from MPM in Fig. 8 agrees with the α-β model, which has been mentioned in Lines 313-314.

3) Comparison of flow velocities from MPM and real measurements: As mentioned in Lines 316-317, the real avalanche with a velocity of 70 m/s was a powder snow avalanche, whose dense core can be captured by the current MPM model while the powder cloud is beyond the scope of this study. We agree that the high velocities (close to 70 m/s) from the real avalanche and the simulated avalanche come from different physical processes. The high velocity of the real avalanche is resulted from the large drop height (1940 m from McClung and Gauer (2018)). In contrast, the high velocity of the simulated avalanche is mainly controlled by the properties (low friction and low cohesion) of the flow. While we observe a generally fair agreement of the MPM and field data in Fig. 9, a quantitative comparison would require full consistency of the model setup (e.g. drop height, flow properties), as we did in Section 4 of the paper. Our main motivation here is to show the influence of mechanical (M and β) and geometrical (θ and $L_0$) properties on the $v_{max}$-α relationship and give a new insight to the negative correlation observed from the data in McClung and Gauer (2018) (Lines 320-324).

According to the relation between the flow velocity and the drop height reported for real snow avalanches (Gauer, 2014), the high flow velocity close to 70 m/s obtained with a drop height of 211.2 m from the MPM simulation might not be realistic for snow avalanches. It has been clarified in Lines 424-426 that the adopted material parameters are designed to study a wide range of different material properties, while the cases with very low friction M and cohesion β leading to the very high velocity might not be realistic for snow avalanches. The material parameters need to be carefully calibrated for investigation of real snow avalanches.

Reply: We tried to plot all the data in one figure for the varying H ($H_0$ in the revised manuscript and hereafter) as shown in Fig. 10 below. The increase of drop height does not necessarily give an increasing maximum velocity if we compare the data with $H_0$ = 73.5 m and the data with $H_0$ = 132.0 m. This is because these two groups do not have the same slope angle in this study. Thus, it is necessary to separately discuss the groups with a fixed slope angle and the groups with a fixed horizontal length, as we did in Figs 8&9. It is mentioned in Lines 133-135 in the revised manuscript that, instead of fixing the horizontal length L ($L_0$ in the revised manuscript) when the slope angle is changed (Groups I, II, III in Table I), one could fix the vertical drop height $H_0$ and change the horizontal length.

[Figure]

Figure 10. Evolution of the maximum velocity with α for varying drop height $H_0$.

*17. Line 355: "Both slope angle and path length have a positive correlation with the maximum front velocity on the slope", this is not that surprise as total Drop height H = L*tan(slope). and Umax \prop f(H).*

Reply: We agree that the total drop height should have a similar effect as the horizontal length L and the slope angle θ. The reason that we separately discuss the slope angle and the path length is that we have both the slope angle and the path length in the calculation of the theoretical maximum 
[revised manuscript text omitted]